# Q-CLIP: Unleashing the Power of Vision-Language Models for Video Quality Assessment through Unified Cross-Modal Adaptation

Yachun Mi [1 2]   Yu Li [1]   Yanting Li [1]   Chen Hui [3]   Tong Zhang [1]   Zhixuan Li [2]   Chenyue Song [1]
Wei Yang Bryan Lim [2]   Shaohui Liu [1]

## Abstract

Accurate and efficient Video Quality Assessment (VQA) has long been a key research challenge. Current mainstream VQA methods typically improve performance by pretraining on large-scale classification datasets, followed by fine-tuning on VQA datasets. However, this strategy presents two significant challenges: (1) merely transferring semantic knowledge learned from pretraining is insufficient for VQA, as video quality depends on multiple factors (e.g., semantics, distortion, motion); (2) pretraining on large-scale datasets demands enormous computational resources, often dozens to hundreds of times more than training on VQA datasets. Recently, Contrastive Vision-Language Models (CVLMs) have shown strong generalization across visual tasks and promising potential for quality assessment. In this work, we propose Q-CLIP, the first fully CVLMs-based framework for VQA. Q-CLIP enhances both visual and textual representations through a Shared Cross-Modal Adapter (SCMA), which contains only a minimal number of trainable parameters and is the only component that requires training. This design significantly reduces computational cost. In addition, we introduce a set of five learnable quality-level prompts to guide the CVLMs in perceiving subtle quality variations. Furthermore, we investigate the impact of different frame sampling strategies on VQA performance. Extensive experiments demonstrate that Q-CLIP exhibits excellent performance on several VQA datasets. *Code:* *https://github.com/xiao-mi-d/Q-CLIP*

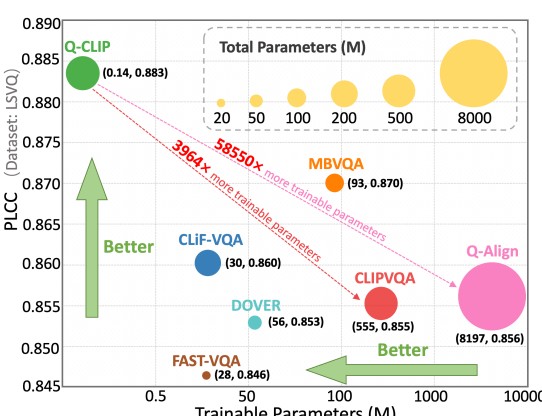

*Figure 1.* Comparison of Q-CLIP with leading VQA methods.

quality videos online. Since video quality directly impacts users' Quality of Experience (QoE), robust Video Quality Assessment (VQA) methods are vital for identifying and filtering subpar content.

Current VQA models fall into two categories: knowledge-driven and data-driven. Knowledge-driven methods (Xu et al., 2014; Saad et al., 2014; Mittal et al., 2015; Korhonen, 2019) depend on handcrafted features, which often fail to capture the complex factors influencing video quality, resulting in limited reliability. In contrast, data-driven methods (Li et al., 2019; Ying et al., 2021; Li et al., 2022; Mi et al., 2025), enabled by subjective VQA datasets, leverage Deep Neural Networks (DNNs) to learn richer representations and achieve superior performance. Nevertheless, the high cost of subjective annotation constrains dataset scale, hindering the full potential of deep learning in VQA.

To address the data scarcity issue, the mainstream solutions adopt a "pretraining-finetuning" paradigm: models are first pre-trained on large-scale classification datasets (e.g., ImageNet (Deng et al., 2009), Kinetics-400 (Kay et al., 2017)), and then fine-tuned on VQA datasets (Hosu et al., 2017; Ying et al., 2021; Wang et al., 2019). While this approach enhances performance, it introduces two critical limitations. First, classification-based pretraining focuses primarily on semantic learning, which only partially captures the perceptual aspects of video quality. Studies (Wu et al., 2022; Li et al., 2019; Wang et al., 2021; Mi et al., 2024b; Yuan et al.,

## 1. Introduction

The proliferation of portable filming devices has made video production more accessible, leading to an influx of low-

[1]School of Computer Science and Technology, Harbin Institute of Technology, Harbin, China [2]College of Computing and Data Science, Nanyang Technological University, Singapore [3]School of Artificial Intelligence, Nanjing University of Information Science and Technology, Nanjing, China. Correspondence to: Shaohui Liu <shliu@hit.edu.cn>.

*Proceedings of the 43[rd] International Conference on Machine Learning*, Seoul, South Korea. PMLR 306, 2026. Copyright 2026 by the author(s).

2024) have shown that video quality depends on multiple dimensions, including semantics, distortion, motion, aesthetics, etc., many of which are not effectively represented by semantic classification alone. Therefore, semantic knowledge learned from classification tasks is inherently limited in its ability to represent overall video quality. Second, the pretraining stage demands substantially more computational resources, often by orders of magnitude compared to fine-tuning. Taking FAST-VQA (Wu et al., 2022) as an example, pretraining on Kinetics-400 is roughly 10× and 200× more expensive than finetuning on LSVQ (Ying et al., 2021) and KoNViD-1k (Hosu et al., 2017), respectively.

Recent advances in Contrastive Vision-Language Models (CVLMs) (Radford et al., 2021; Jia et al., 2021; Zhai et al., 2023; Xu et al., 2024; Bolya et al., 2025) have introduced new perspectives for solving visual tasks. Trained on large-scale image–text pairs, these models acquire rich multi-modal knowledge and demonstrate impressive generalization capabilities across domains (Zhang et al., 2024a). Unlike traditional classification-based pretraining, which primarily emphasizes semantic discrimination, CVLMs inherently encode cross-modal representations that capture the multifaceted nature of video quality, including perceptual distortions (e.g., blur, noise), motion dynamics, aesthetic preferences, and semantic consistency. Moreover, recent studies (Mi et al., 2024b;a; 2026; Wang et al., 2023a) show that CVLMs perform well in zero-shot quality prediction across multiple quality-related dimensions, even without task-specific supervision, highlighting their strong potential in quality perception. These properties suggest that CVLMs may serve as a promising alternative to classification-based pretraining strategies, offering a more holistic understanding of video quality while also alleviating the computational burden associated with large-scale pretraining.

Despite their strong generalization, adapting CVLMs to VQA remains challenging. The performance of CVLMs in downstream tasks is often constrained by limited intra- and cross-modal representational capacity, particularly in fine-grained perceptual tasks (Liang et al., 2022; Qian et al., 2023; Zhang et al., 2023a). However, as a typical fine-grained perceptual task, VQA requires the model to capture subtle quality differences and rely heavily on localized visual cues, which amplifies the challenge of transferring knowledge from CVLMs. Moreover, fine-tuning CVLMs for VQA not only incurs substantial computational costs but also risks degrading their original representational capabilities. Given these difficulties, all VQA methods (Mi et al., 2024b; Xing et al., 2025; Yuan et al., 2024) that incorporate CVLMs employ CVLMs only as auxiliary feature extractors combined with backbone networks, not as standalone frameworks (Appendix. A). This study aims to explore how to enhance CVLMs' perception of quality-related factors while minimizing computational overhead, thereby constructing a VQA framework entirely based on CVLMs.

In addition, fine-grained prompts play a crucial role in providing textual guidance for CVLMs (Zhou et al., 2022a;b; Lu et al., 2022; Ju et al., 2022; Yang et al., 2024). Existing CVLMs-utilizing quality assessment methods (Wu et al., 2023c; Wang et al., 2023a) often rely on antonym pairs (e.g., "good" and "bad") to guide quality perception. However, such binary prompts provide only coarse-grained supervision and may be insufficient for capturing the full aspects for describing the video quality (Mi et al., 2024b;a). Recent studies (Wu et al., 2024; You et al., 2025) in Multimodal Large Language Models (MLLMs)-based quality assessment show that mapping quality scores to a five-level scale (excellent, good, fair, poor, bad) leads to more accurate predictions. Accordingly, we consider similar prompt strategies as a promising direction for enhancing CVLMs in VQA.

Based on the above analysis, we introduce Q-CLIP, a VQA method based entirely on CVLMs. Specifically, we design a Shared Cross-Modal Adapter (SCMA) to enhance the representations of the visual and textual branches. This adapter consists of only a few fully connected layers (**0.14M**) and is the only component that requires training, significantly reducing computational overhead. As shown in Fig. 1, Q-CLIP achieves the best performance while training only a minimal number of parameters. In addition, we develop a set of learnable five-level prompts to provide fine-grained textual quality descriptions as input guidance to the CVLMs. This allows us to jointly consider the similarity scores between the video and prompts of different quality levels, enabling more accurate quality prediction. Furthermore, we investigate the impact of different frame sampling strategies on VQA performance. Previous works (Wu et al., 2022; Wen et al., 2024) mainly adopt random or uniform sampling, with limited exploration of its impact on VQA performance. Beyond conventional methods, we explore frame-difference-based sampling strategies to assess its potential benefits for VQA. Our findings offer new insights that may inform and inspire future research in this direction.

Our contributions can be summarized as follows:

- We introduce Q-CLIP, a fully CVLMs-based VQA model. By incorporating an extremely lightweight adapter, Q-CLIP effectively boosts the VQA capabilities of CVLMs at a remarkably low training cost.

- We design a learnable five-level prompt mechanism to guide CVLMs in perceiving subtle quality variations.

- This work presents a systematic study of frame sampling strategies, offering new insights and practical guidance for future research in VQA.

- Extensive experiments demonstrate that Q-CLIP achieves state-of-the-art performance across multiple VQA datasets.

## 2. Related Work

### 2.1. VQA methods

**Knowledge-driven.** Knowledge-driven methods (Mittal et al., 2015; 2012; Tu et al., 2021a; Korhonen, 2019; Tu et al., 2021b; Xu et al., 2014; Saad et al., 2014) assess video quality by extracting handcrafted features. For example, VIIDEO (Mittal et al., 2015) utilizes intrinsic statistical regularities of natural videos to capture anomalous information caused by distortion. TLVQM (Korhonen, 2019) extracts low-complexity motion features and high-complexity spatial features. VIDEAL (Tu et al., 2021a) detects and quantifies distortions by extracting a diverse set of perceptual quality features. However, handcrafted features struggle to capture the complex and diverse factors affecting video quality, leading to suboptimal performance.

**Data-driven.** Data-driven methods automatically extract quality-aware features by training DNNs on high-quality VQA datasets. For example, GST-VQA (Chen et al., 2021) and VSFA (Li et al., 2019) use pretrained 2D Convolutional Neural Networks (CNNs) (Simonyan & Zisserman, 2014; He et al., 2016) combined with GRU (Cho et al., 2014) for spatiotemporal modeling, while other studies (Ying et al., 2021; Li et al., 2022; Sun et al., 2022; Wang et al., 2021; Zhang et al., 2023b; Wen et al., 2024) further incorporate 3D-CNNs (Tran et al., 2015; Hara et al., 2018; 2017) to enhance spatiotemporal feature extraction. In addition, Transformer-based VQA (Wu et al., 2023b; 2022; 2023a;e; Liu et al., 2023) is gradually gaining more competitive performance. For example, FAST-VQA (Wu et al., 2022) and FasterVQA (Wu et al., 2023a) sample spatial-temporal grids and utilize modified Video Swin Transformers (Liu et al., 2022). However, these fragment sampling strategies often neglect semantic content. Based on this, DOVER (Wu et al., 2023d) and Zoom-VQA (Zhao et al., 2023) further introduce a semantic branch to enhance FAST-VQA. With the success of CVLMs (Radford et al., 2021; Tschannen et al., 2025; Bolya et al., 2025), applying them to VQA has become a growing research focus. CLiF-VQA (Mi et al., 2024b) and PTM-VQA (Yuan et al., 2024) extract human feeling features from CLIP (Radford et al., 2021) under language supervision, serving as a complement to spatiotemporal features. Similarly, MaxVQA (Wu et al., 2023c) and CLIPVQA (Xing et al., 2025) integrate CLIP with backbone networks to extract multimodal features for VQA.

### 2.2. Contrastive Vision-Language Models

Contrastive Vision-Language Models (CVLMs) are trained with contrastive learning on large-scale image–text pairs to align visual and textual representations in a shared embedding space. CLIP (Radford et al., 2021) is a representative model known for its robust representations and strong generalization, learned from comparative training on 400 million image-text pairs. Subsequently, a series of enhanced versions of CLIP are proposed. For example, MetaCLIP (Xu et al., 2024) enhances the performance of CLIP by curating and balancing the raw data from the network to improve the quality of the training data. SigLIP (Zhai et al., 2023) replaces the original Softmax with Sigmoid when calculating the loss, leading to improved computational efficiency as well as better performance. Furthermore, SigLIP2 (Tschannen et al., 2025) is trained on a larger-scale dataset and unifies previously disjoint training strategies into a structured, multi-stage pipeline, resulting in notable performance improvements. Recently, (Bolya et al., 2025) train CLIP on a larger image dataset and fine-tune it on a 22M-sized video dataset, significantly enhancing its generalization to video data. For example, CoOp (Zhou et al., 2022a), CLIP-Adapter (Gao et al., 2024), and Tip-Adapter (Zhang et al., 2021) enhance CVLMs to enable few-shot image recognition. Moreover, some works extend CVLMs to video tasks. VideoCLIP (Xu et al., 2021) replaces image-text pairs with video-text pairs for video understanding. CLIP4Clip (Luo et al., 2022) adapts CLIP for video retrieval by fine-tuning it end-to-end. ActionCLIP (Wang et al., 2023b) and XCLIP (Ni et al., 2022) directly transfer CLIP's visual representations to video recognition.

## 3. Proposed Method

### 3.1. Overall Architecture

Our proposed Q-CLIP architecture, illustrated in Fig. 2, is fully built upon the CVLMs framework. It enhances the performance of CVLMs in VQA by introducing two novel modules: Shared Cross-Modal Adapter (SCMA) and a set of learnable five-level quality prompts. In addition, we investigate the impact of different frame sampling strategies on VQA performance. Beyond conventional random and uniform sampling, we explore a motion-based approach that calculates the difference between each frame and its adjacent frames to measure the intensity of motion. Frames are then sampled according to predefined rules based on these motion differences.

### 3.2. Shared Cross-Modal Adapter

We propose a Shared Cross-Modal Adapter (SCMA) to efficiently adapt frozen CVLMs to VQA, as shown in Fig. 3. SCMA refines both visual and textual representations via a shared adaptation mechanism, enabling the model to learn task-specific updates without introducing modality-dependent rules. Moreover, by sharing the adapter core across encoder layers, SCMA avoids parameter growth with depth, which not only preserves its extremely lightweight training characteristic but also reduces the risk of overfitting.

Let $m \in \{v, t\}$ denote the modality (visual/text). In typical

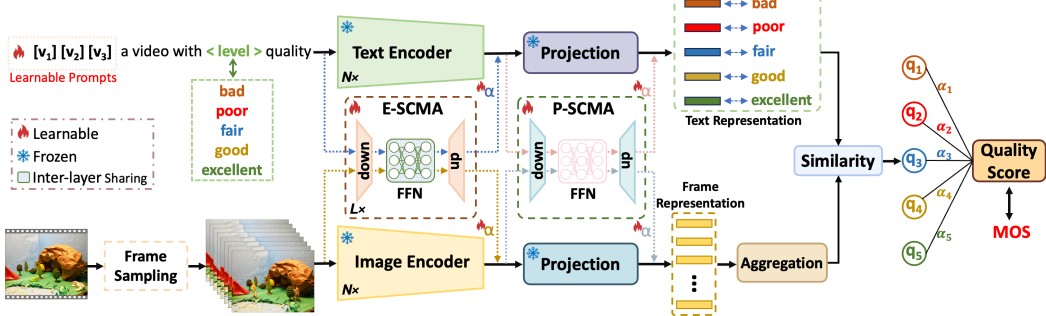

*Figure 2.* The overall framework of the proposed Q-CLIP.

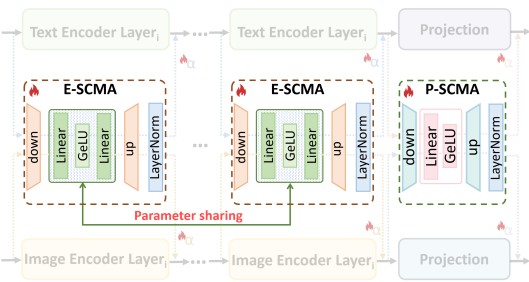

*Figure 3.* Architecture of the proposed SCMA.

CVLMs architectures, the visual and textual branches share the same hidden width $d$. At encoder layer $k$, we denote the token features as $\mathbf{X}_k^m \in \mathbb{R}^{B \times L_m \times d}$, where $B$ is the batch size and $L_m$ is the sequence length. A frozen modality-specific encoder block produces:

$$\mathbf{H}_k^m = E_k^m(\mathbf{X}_k^m), \qquad m \in \{v, t\} \tag{1}$$

**SCMA formulation.** SCMA follows a residual bottleneck structure. At layer $k$, it first projects $\mathbf{H}_k^m$ to a low-dimensional bottleneck of size $r \ll d$, applies a shared core transform, and then projects back to the hidden space:

$$\mathbf{Z}_k^m = \mathbf{D}_k \, \mathbf{H}_k^m, \qquad \mathbf{D}_k \in \mathbb{R}^{d \times r} \tag{2}$$

$$\tilde{\mathbf{Z}}_k^m = f_\theta(\mathbf{Z}_k^m), \qquad f_\theta : \mathbb{R}^r \to \mathbb{R}^r \tag{3}$$

$$\mathbf{A}_k^m = \mathrm{LN}\left(\mathbf{U}_k \, \tilde{\mathbf{Z}}_k^m\right), \qquad \mathbf{U}_k \in \mathbb{R}^{r \times d} \tag{4}$$

$$\mathbf{X}_{k+1}^m = \mathbf{H}_k^m + \gamma_k^m \, \mathbf{A}_k^m, \qquad \gamma_k^m \in \mathbb{R} \tag{5}$$

Here $\mathrm{LN}(\cdot)$ denotes LayerNorm, and $\gamma_k^m$ is a learnable residual scaling factor that stabilizes training and controls the update magnitude. The bottleneck dimension $r$ is chosen to be small so that SCMA introduces only a negligible number of trainable parameters.

**Cross-modal sharing and inter-layer sharing.** Since the two branches share the same hidden width $d$, we share the bottleneck projections across modalities within each layer, i.e., the visual and textual branches use the same $(\mathbf{D}_k, \mathbf{U}_k)$ at layer $k$:

$$\mathbf{D}_k^v = \mathbf{D}_k^t \triangleq \mathbf{D}_k, \qquad \mathbf{U}_k^v = \mathbf{U}_k^t \triangleq \mathbf{U}_k \tag{6}$$

This sharing enforces a unified adaptation rule for both branches, which stabilizes optimization under a frozen backbone and avoids branch-specific update heuristics.

To control the adapter capacity as depth increases and mitigate overfitting, we share the core transform $f_\theta$ across all encoder layers, so that the number of core parameters does not scale with depth:

$$f_\theta^{(1)} \equiv f_\theta^{(2)} \equiv \cdots \equiv f_\theta^{(K)} \equiv f_\theta \tag{7}$$

Consequently, $\{\mathbf{D}_k, \mathbf{U}_k\}$ are layer-specific and modality-shared, while $f_\theta$ is shared across modalities and layers.

**Instantiations.** We instantiate SCMA in two locations. For the encoder-side adapter (E-SCMA), we parameterize $f_\theta$ as a compact nonlinear mapping to enhance representational flexibility while maintaining a small parameter budget. For the projection-side adapter (P-SCMA), we use a linear instantiation of the core transform to match the linear nature of similarity projection. Both instantiations follow the same bottleneck and sharing schemes in Eqs. (2)–(7), differing only in the choice of $f_\theta$. Further analysis in Appendix G.

### 3.3. Learnable Five-level Prompts

Using antonym-based prompts in CVLMs has shown promising results for quality perception. However, since quality assessment is a fine-grained prediction task, such binary prompts are overly coarse and may limit the performance of CVLMs in capturing subtle quality differences. Fortunately, recent studies on quality perception using MLLMs have demonstrated that converting quality scores into discrete quality levels helps models better capture nuanced hierarchical patterns, leading to improved performance. Inspired by this, we argue that a similar design is also beneficial for CVLMs. Therefore, we introduce a prompt scheme with five distinct quality levels:

$$p = \text{``a video of''} + < level > + \text{``quality''} \tag{8}$$

Here, $< level >$ represents the five quality levels: excellent, good, fair, poor, bad. However, more specific prompts can often introduce bias into the perception of CVLMs. To

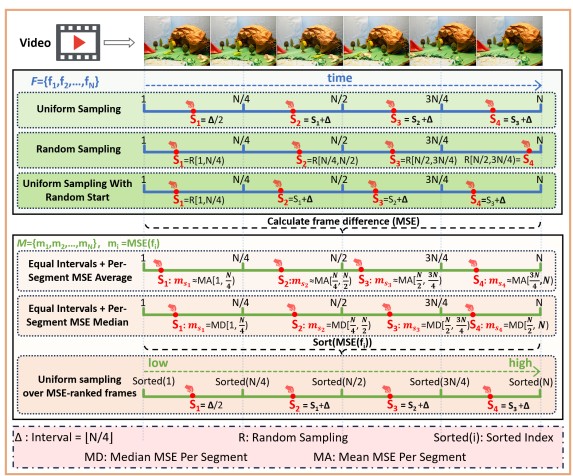

*Figure 4.* Frame Sampling Diagram.

address this, we introduce learnable prompts to optimize the initial five-level prompt scheme:

$$\hat{p} = Learnable(\text{"X X X"}) + p \quad (9)$$

We initialize the prompts with three "X" tokens and optimize them during training. All other parts of prompts remain frozen. The prompts can be formulated as:

$$P = \{\hat{p}_{exc}, \hat{p}_{good}, \hat{p}_{fair}, \hat{p}_{poor}, \hat{p}_{bad}\} \quad (10)$$

### 3.4. Quality Regression

The video and text prompts are processed by the model to obtain video features $V$ and text features $T = \{t_{exc}, t_{good}, t_{fair}, t_{poor}, t_{bad}\}$, respectively. Then, calculate the cosine similarity between the visual content and prompts to predict the score for each dimension:

$$s_k = \frac{t_k \cdot V}{\|t_k\| \|V\|}, k \in \{exc, good, fair, poor, bad\} \quad (11)$$

These scores $S = \{s_{exc}, s_{good}, s_{fair}, s_{poor}, s_{bad}\}$ form a quality-level-related distribution, which is further processed to generate the final quality assessment score. Finally, by applying a weighted sum, the discrete similarity scores are converted into a continuous quality prediction:

$$Q_{pred} = \sum_{k=exc}^{bad} w_k \cdot s_k \quad (12)$$

where $w_k$ are learnable weights adjusting each quality level's contribution to the final prediction.

### 3.5. Frame-Difference-Based Sampling

VQA relies heavily on representative frame samples, as full video sequences are often computationally prohibitive and redundant. While random and uniform sampling are widely used as baselines, they overlook the dynamic characteristics

of videos that may correlate with quality perception (e.g., motion intensity). To address this, we systematically investigate the impact of frame sampling strategies on VQA performance, with a particular focus on frame-difference-based sampling, a strategy rarely explored in prior VQA literature. Frame differences, quantified via pixel-wise MSE, reflect motion intensity between consecutive frames:

$$m_t = \frac{1}{2}MSE(v_i, v_{i+1}) + MSE(v_i, v_{i-1}) \quad (13)$$

where $MSE(a,b) = \frac{1}{H \times W \times 3}\sum_p (a_p - b_p)^2$ computes the pixel-wise MSE between frames $a$ and $b$ with ($p$ indexing individual pixels). The first and last frames are compared only with their single adjacent frames. Using the frame difference MSE $\{m_1, m_2, ...m_N\}$, we design three sampling strategies to select frames, as illustrated in Fig. 4, which shows the detailed sampling process. Additional details regarding the sampling process can be found in Appendix. E.

## 4. Experiments

### 4.1. Experimental Setups

**Datasets.** We verify our model on six datasets: LSVQ (Ying et al., 2021), KoNViD-1k (1200) (Hosu et al., 2017), LIVE-VQC (585) (Sinno & Bovik, 2018), YouTube-UGC (1067) (Wang et al., 2019), CVD2014 (234) (Nuutinen et al., 2016), LIVE-Qualcomm (208) (Ghadiyaram et al., 2017). We pre-train Q-CLIP on LSVQ (28056), with intra-dataset testing on LSVQ$_{test}$ (7400) and LSVQ$_{1080p}$ (3600), and cross-dataset testing on KoNViD-1k and LIVE-VQC. Further, we fine-tune the model on KoNViD-1k, LIVE-VQC, YouTube-UGC, CVD2014 and LIVE-Qualcomm. Following standard practice (Wu et al., 2022; Mi et al., 2024b; Wu et al., 2023a), we split each dataset into 80% training and 20% testing.

**Evaluation Criteria**. The Spearman Rank Order Correlation Coefficient (SRCC), the Kendall Rank Order Correlation Coefficient (KRCC), the Pearson Linear Correlation Coefficient (PLCC), and the Root Mean Square Error (RMSE) are used as evaluation metrics.

**Implementation Details.** We employ PyTorch framework and an NVIDIA GeForce RTX 4090 card to train the model in all experimental implementations. As most current CVLMs are trained on static images, they are not well-equipped to model the temporal dynamics in videos. To address this, we adopt a CLIP variant (Bolya et al., 2025) that has been pre-tuned on video data as our backbone. We sample 8 frames per video as input. We default to mean pooling for frame aggregation, supported by our experimental results (See Appendix. F.3). We set the initial learning rate to 0.001, the optimizer to AdamW, and use a cosine annealing strategy to dynamically adjust the learning rate. And training is conducted for 8 epochs using a batch size of 12. More experimental details are in Appendix. D.

*Table 1.* Experimental performance of the pre-trained Q-CLIP on LSVQ. The best and second-best results are **bolded** and underlined.

| Testing Type | | | Intra-dataset Test Datasets | | | | Cross-dataset Test Datasets | | | |
|---|---|---|---|---|---|---|---|---|---|---|
| Testing Datasets | | | **LSVQ$_{test}$** | | **LSVQ$_{1080p}$** | | **KoNViD-1k** | | **LIVE-VQC** | |
| Type | Methods | Source | SRCC↑ | PLCC↑ | SRCC↑ | PLCC↑ | SRCC↑ | PLCC↑ | SRCC↑ | PLCC↑ |
| Knowledge-driven | BRISQUE | *TIP, 2012* | 0.569 | 0.576 | 0.497 | 0.531 | 0.646 | 0.647 | 0.524 | 0.536 |
| | TLVQM | *TIP, 2019* | 0.772 | 0.774 | 0.589 | 0.616 | 0.732 | 0.724 | 0.670 | 0.691 |
| | VIDEVAL | *TIP, 2021* | 0.794 | 0.783 | 0.545 | 0.554 | 0.751 | 0.741 | 0.630 | 0.640 |
| Data-driven | VSFA | *ACMMM, 2019* | 0.801 | 0.796 | 0.675 | 0.704 | 0.784 | 0.794 | 0.734 | 0.772 |
| | PVQ | *CVPR, 2021* | 0.827 | 0.828 | 0.711 | 0.739 | 0.791 | 0.795 | 0.770 | 0.807 |
| | BVQA | *TCSVT, 2022* | 0.852 | 0.854 | 0.771 | 0.782 | 0.834 | 0.837 | 0.816 | 0.824 |
| | FAST-VQA | *ECCV, 2022* | 0.876 | 0.877 | 0.779 | 0.814 | 0.859 | 0.855 | 0.823 | 0.844 |
| | DOVER | *ICCV, 2023* | 0.881 | 0.879 | 0.782 | 0.827 | 0.871 | 0.872 | 0.812 | 0.841 |
| | Zoom-VQA | *CVPR, 2023* | 0.886 | 0.879 | 0.799 | 0.819 | 0.877 | 0.875 | 0.814 | 0.833 |
| | MBVQA | *CVPR, 2024* | 0.895 | 0.895 | 0.809 | 0.844 | 0.878 | 0.884 | 0.806 | 0.844 |
| MLLMs | Q-Align | *ICML, 2024* | 0.883 | 0.882 | 0.797 | 0.830 | 0.865 | 0.877 | NA | NA |
| CVLMs | PTM-VQA | *CVPR, 2024* | 0.855 | 0.864 | 0.736 | 0.782 | 0.824 | 0.830 | 0.785 | 0.737 |
| | CLiF-VQA | *ACMMM, 2024* | 0.886 | 0.887 | 0.790 | 0.832 | 0.877 | 0.874 | **0.834** | 0.855 |
| | CLIPVQA | *TBC, 2025* | 0.881 | 0.883 | 0.782 | 0.827 | 0.864 | 0.887 | 0.781 | **0.871** |
| | **Q-CLIP** *-RandSampl* | | 0.895 | 0.896 | 0.814 | 0.852 | 0.882 | 0.892 | 0.808 | 0.843 |
| | **Q-CLIP** *-UNISampl* | | 0.897 | 0.895 | 0.820 | 0.853 | 0.883 | 0.891 | 0.803 | 0.842 |
| | **Q-CLIP** *-UNIRandStart* | | 0.893 | 0.895 | 0.818 | 0.858 | 0.883 | 0.890 | 0.804 | 0.844 |
| | **Q-CLIP** *-MSESortedUNI* | | 0.891 | 0.893 | 0.812 | 0.852 | 0.888 | 0.894 | 0.810 | 0.845 |
| | **Q-CLIP** *-SegMSEMean* | | 0.897 | 0.896 | 0.820 | 0.855 | 0.889 | 0.895 | 0.813 | 0.851 |
| | **Q-CLIP** *-SegMSEMedian* | | 0.891 | 0.893 | 0.813 | 0.852 | 0.889 | 0.896 | 0.813 | 0.852 |
| | **Q-CLIP** *-Mixed* | | **0.899** | **0.900** | **0.823** | **0.866** | **0.896** | **0.901** | 0.826 | 0.867 |

*Table 2.* The finetune results on multiple small datasets. The best and second-best results are **bolded** and underlined.

| Finetune Datasets | | | **LIVE-VQC** | | **KoNViD-1k** | | **YouTube-UGC** | | **CVD2014** | | **LIVE-Qualcomm** | |
|---|---|---|---|---|---|---|---|---|---|---|---|---|
| Type | Methods | Source | SRCC↑ | PLCC↑ | SRCC↑ | PLCC↑ | SRCC↑ | PLCC↑ | SRCC↑ | PLCC↑ | SRCC↑ | PLCC↑ |
| Knowledge-driven | TLVQM | *TIP, 2019* | 0.799 | 0.803 | 0.773 | 0.768 | 0.669 | 0.659 | 0.830 | 0.850 | 0.770 | 0.810 |
| | VIDEVAL | *TIP, 2021* | 0.752 | 0.751 | 0.783 | 0.780 | 0.779 | 0.773 | NA | NA | NA | NA |
| | RAPIQUE | *OJSP, 2021* | 0.755 | 0.786 | 0.803 | 0.817 | 0.759 | 0.768 | NA | NA | NA | NA |
| Data-driven | VSFA | *ACMMM, 2019* | 0.773 | 0.795 | 0.773 | 0.775 | 0.724 | 0.743 | 0.870 | 0.868 | 0.737 | 0.732 |
| | GST-VQA | *TCSVT, 2021* | NA | NA | 0.814 | 0.825 | NA | NA | 0.831 | 0.844 | 0.801 | 0.825 |
| | PVQ | *CVPR, 2021* | 0.827 | 0.837 | 0.791 | 0.786 | NA | NA | NA | NA | NA | NA |
| | BVQA | *TCSVT, 2022* | 0.841 | 0.839 | 0.835 | 0.834 | 0.825 | 0.818 | 0.863 | 0.883 | 0.833 | 0.837 |
| | FAST-VQA | *ECCV, 2022* | 0.845 | 0.852 | 0.890 | 0.889 | 0.857 | 0.853 | 0.891 | 0.903 | 0.819 | 0.851 |
| | DOVER | *ICCV, 2023* | 0.812 | 0.852 | 0.897 | 0.899 | 0.877 | 0.873 | 0.858 | 0.881 | 0.736 | 0.789 |
| | MBVQA | *CVPR, 2024* | 0.860 | 0.880 | 0.901 | 0.905 | 0.876 | 0.877 | 0.883 | 0.901 | 0.832 | 0.842 |
| CVLMs | MaxVQA | *ACMMM, 2023* | 0.854 | 0.873 | 0.894 | 0.895 | 0.894 | 0.890 | NA | NA | NA | NA |
| | PTM-VQA | *CVPR, 2024* | 0.811 | 0.820 | 0.857 | 0.872 | 0.858 | 0.857 | NA | NA | NA | NA |
| | CLiF-VQA | *ACMMM, 2024* | 0.866 | 0.878 | 0.903 | 0.903 | 0.888 | 0.890 | 0.881 | 0.891 | 0.832 | 0.850 |
| | CLIPVQA | *TBC, 2025* | 0.870 | 0.892 | 0.907 | 0.912 | 0.881 | 0.883 | 0.883 | 0.888 | 0.833 | 0.872 |
| | **Q-CLIP** | *Ours* | **0.881** | **0.901** | **0.915** | **0.920** | **0.911** | **0.911** | **0.897** | **0.907** | **0.846** | **0.884** |

## 4.2. Pre-training Results on LSVQ

We pre-train the proposed Q-CLIP on LSVQ and conduct intra-dataset testing on LSVQ$_{test}$ and LSVQ$_{1080p}$. Additionally, cross-dataset testing performed on KoNViD-1k and LIVE-VQC. Furthermore, we examine the effectiveness of various frame sampling methods. The results are shown in Tab. 1. Frame-difference-based samplings achieve comparable results to traditional methods, such as random and uniform sampling, in intra-dataset testing. In cross-dataset testing, frame-difference-based samplings demonstrate superior performance compared to traditional approaches, indicating better generalization capability. This suggests that frame-difference-based samplings can more effectively select frames that are representative of video quality, particularly in cross-dataset scenarios. In contrast, traditional

*Table 3.* Comparison of different fine-tuning methods on LSVQ.

| Methods | SRCC↑ | PLCC↑ |
|---|---|---|
| Full fine-tuning | 0.816 | 0.811 |
| CoOp | 0.763 | 0.764 |
| VPT | 0.823 | 0.820 |
| CLIP-Adapter | 0.881 | 0.884 |
| LoRA | 0.883 | 0.883 |
| **Ours** | **0.897** | **0.895** |

sampling methods do not consider any intrinsic characteristics of the video frames, which may result in redundant or less informative samples, thereby limiting their representativeness and generalizability. The results indicate that regardless of the sampling strategy employed, Q-CLIP consistently achieves state-of-the-art performance. Moreover, integrating multiple sampling methods during training further enhances the model's overall performance.

*Table 4.* Ablation on SCMA.

| Visual | Text | Sharing | Layer sharing | SRCC↑ | PLCC↑ |
|--------|------|---------|---------------|-------|-------|
| ✓ | | | | 0.866 | 0.864 |
| | ✓ | | | 0.837 | 0.838 |
| ✓ | ✓ | | | 0.875 | 0.878 |
| ✓ | ✓ | ✓ | | 0.885 | 0.886 |
| ✓ | ✓ | ✓ | ✓ | **0.895** | **0.897** |

*Table 5.* Ablation on prompts. * : learnable.

| Datasets | $LSVQ_{test}$ | | KoNViD-1k | | LIVE-VQC | |
|----------|-------|-------|-------|-------|-------|-------|
| Prompts | SRCC↑ | PLCC↑ | SRCC↑ | PLCC↑ | SRCC↑ | PLCC↑ |
| *Antonym* | 0.883 | 0.881 | 0.866 | 0.867 | 0.791 | 0.820 |
| *Antonym\** | 0.885 | 0.883 | 0.871 | 0.879 | 0.789 | 0.826 |
| *Five levels* | 0.891 | 0.891 | 0.874 | 0.885 | 0.798 | 0.837 |
| *Five levels\** | **0.895** | **0.897** | **0.883** | **0.891** | **0.803** | **0.842** |

et al., 2022), CLIP-Adapter (Gao et al., 2024), and LoRA (Hu et al., 2022), as shown in Tab. 3. Full fine-tuning of CLIP does not yield satisfactory performance. This is primarily because updating all parameters not only disrupts the well-learned representations of CLIP, but also suffers from insufficient training data, making it difficult to effectively optimize the entire model. Due to CoOp optimizing only the prompt and VPT optimizing only the visual branch, both methods perform poorly. In contrast, CLIP-Adapter and LoRA yield more competitive results by preserving the pre-trained knowledge of CLIP and enabling effective adaptation. Nevertheless, their performance still falls short of our proposed SCMA, which demonstrates superior effectiveness in the VQA task. Additional analysis is provided in the Appendix. C.

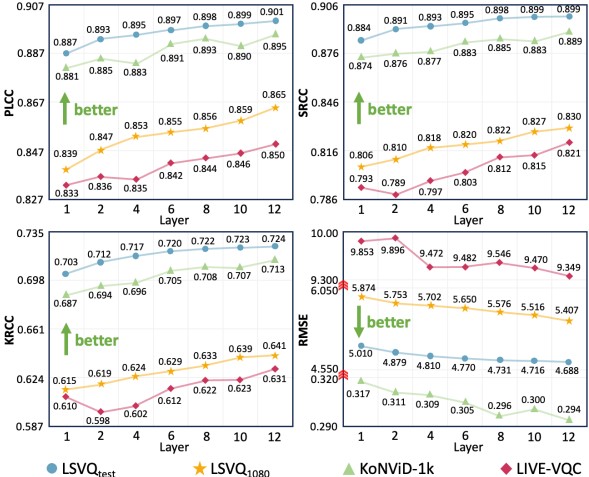

*Figure 5.* Ablation on the number of E-SCMA Layers.

Compared with knowledge-driven and data-driven methods, Q-CLIP achieves a significant improvement over all datasets. Furthermore, it demonstrates distinct advantages over Q-Align, which is based on MLLMs. Compared with CVLMs-utilizing methods, although Q-CLIP performs slightly lower than CLiF-VQA and CLIPVQA on LIVE-VQC, it significantly outperforms them on the other three datasets. Specifically, Q-CLIP improves over CLiF-VQA and CLIPVQA by up to **3.7%** in SRCC and **2.9%** in PLCC.

### 4.3. Fine-tuning Results on Small Datasets

After pre-training on LSVQ, we fine-tune Q-CLIP on five small datasets (LIVE-VQC, KoNViD-1k, YouTube-UGC, CVD2014, LIVE-Qualcomm), as shown in Tab. 2. Specifically, we use uniform sampling as the sampling strategy during fine-tuning. As can be seen, Q-CLIP achieves unprecedented performance on all five datasets. Compared to the current best performance, Q-CLIP improves the average performance on SRCC and PLCC by **1.39%** and **1.22%**, respectively. Furthermore, Q-CLIP outperforms the state-of-the-art CVLMs-utilizing method CLIPVQA by **1.74%** and **1.72%** in SRCC and PLCC, respectively. The results further illustrate the validity of Q-CLIP.

### 4.4. Comparison with Other Fine-Tuning Methods

To validate the effectiveness of SCMA, we compare it against several mainstream fine-tuning approaches, including full fine-tuning, CoOp (Zhou et al., 2022c), VPT (Jia

### 4.5. Ablation Studies

We conduct experimental analysis to evaluate the effectiveness of each component. Ablation experiments are by default based on a uniform sampling strategy. See Appendix. F for more ablation experiments.

**Ablation on SCMA.** We validate the effectiveness of SCMA on LSVQ, as shown in Tab. 4. Applying SCMA to either the visual or textual branch individually results in limited performance. When SCMA is applied to both branches without parameter sharing, the performance improves notably. Sharing SCMA across the two branches leads to further gains, and the best results are achieved when inter-layer sharing is additionally introduced. These results validate the effectiveness of our proposed SCMA architecture, which jointly leverages branch-wise and inter-layer sharing.

Furthermore, since CVLMs typically consist of multiple layers, we further investigate the impact of the number of inserted E-SCMA layers on model performance. As shown in Fig. 5, we train the model on $LSVQ_{train}$ and evaluate it on $LSVQ_{test}$, $LSVQ_{1080p}$, KoNViD-1k, and LIVE-VQC. The results demonstrate that as the number of E-SCMA layers increases, the model performance consistently improves. Notably, all comparative experiments in this paper are based on a 6-layer E-SCMA configuration, suggesting that further performance gains can be achieved by increasing the number of E-SCMA layers.

**Ablation on Prompts.** Most existing CVLMs-utilizing quality assessment methods (Wu et al., 2023c; Wang et al., 2023a) utilize antonym-based prompts. To validate the ef-

*Table 6.* Ablation on backbone.

| Datasets | LSVQ$_{test}$ | | LSVQ$_{1080p}$ | | KoNViD-1k | |
|---|---|---|---|---|---|---|
| Backbone | SRCC↑ | PLCC↑ | SRCC↑ | PLCC↑ | SRCC↑ | PLCC↑ |
| CLIP-B/16 | 0.885 | 0.887 | 0.799 | 0.842 | 0.881 | 0.878 |
| CLIP-B/32 | 0.887 | 0.887 | 0.797 | 0.843 | 0.879 | 0.879 |
| CLIP-L/14 | 0.893 | 0.894 | 0.814 | 0.852 | 0.885 | 0.892 |
| **Q-CLIP** | **0.899** | **0.900** | **0.823** | **0.866** | **0.896** | **0.901** |

fectiveness of our proposed prompts, we compare it against the antonym-based prompts, as shown in Tab. 5. Compared to antonym-based prompts, our five-level prompt design offers a clear advantage. Furthermore, performance is further improved by introducing learnable parameters.

**Ablation on Backbone.** To further validate that Q-CLIP's performance does not entirely depend on a specific CLIP backbone network fine-tuned on the video. We instantiate Q-CLIP with three standard image-pretrained CLIP backbones (Radford et al., 2021): CLIP-B/16, CLIP-B/32, and CLIP-L/14. Specifically, for each backbone, we employ the same experimental setup as Q-CLIP. We pre-train the proposed Q-CLIP on LSVQ and conduct intra-dataset testing on LSVQ$_{test}$ and LSVQ$_{1080p}$. Additionally, cross-dataset testing is performed on KoNViD-1k and LIVE-VQC. Detailed results are shown in Tab. 6. The results demonstrate that, as we progressively move from the smallest image-based backbones (CLIP-B/16 and CLIP-B/32) to slightly larger CLIP-L/14 and Q-CLIP, performance remains consistently strong. The Q-CLIP with a video-tuned backbone brings only a marginal improvement on top of this, indicating that the smaller image-pretrained CLIP backbones are already sufficient for Q-CLIP to achieve SOTA performance while offering a better efficiency–accuracy trade-off.

### 4.6. Efficiency

Compared to models specifically designed for VQA, CVLMs typically have a larger number of parameters, making model efficiency a critical consideration. To this end, we conduct extensive experiments to evaluate the efficiency of Q-CLIP. We first compare the total and fine-tuned parameters with several state-of-the-art VQA methods, as shown in Fig. 1. Q-CLIP requires only a minimal number of fine-tuned parameters (**0.14M**) to achieve top performance. For example, compared with MBVQA, CLIP-VQA, and Q-Align, Q-CLIP reduces the fine-tuned parameters by **664×**, **3964×**, and **58550×** respectively. Moreover, even when compared to the most efficient method, FAST-VQA, Q-CLIP still achieves a **200×** reduction in fine-tuned parameters. Additionally, Q-CLIP's total parameters remain reasonable. Compared to CLIPVQA, which is also based on CVLMs, Q-CLIP has a comparable total parameter size. In contrast, Q-CLIP reduces the total parameters by **13×** compared to Q-Align, which is based on MLLMs. See Appendix. B for additional efficiency experiments.

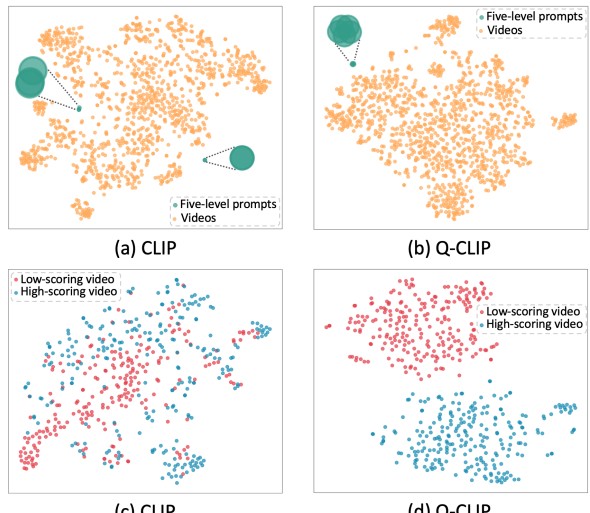

*Figure 6.* t-SNE visualizations on KoNViD-1k dataset.

### 4.7. Visualization

To further evaluate Q-CLIP's quality perception capability, we visualize feature distributions using t-SNE (Fig. 6). From Fig. 6a and Fig. 6b, the original CLIP shows clear modality confusion, as visual and textual features overlap significantly. Additionally, the five-level prompt features are scattered and poorly separated, with some quality levels fully overlapping. This indicates that CLIP fails to effectively encode quality-relevant information. In contrast, Q-CLIP exhibits a well-structured feature space, where visual and textual modalities are distinctly separated, and quality levels form compact, aligned clusters. Such separability is key to robust cross-modal understanding in high-performing CVLMs (Liang et al., 2022; Qian et al., 2023; Zhang et al., 2023a). Further analysis of video features by quality (Fig. 6c and Fig. 6d) shows that Q-CLIP clearly distinguishes high- and low-scoring videos, unlike CLIP, which presents significant overlap. These results confirm that Q-CLIP enhances the feature space through effective modality separation and structured quality encoding, enabling accurate quality perception. See Appendix. C for more visualization analysis.

### 5. Conclusion

In this paper, we introduce Q-CLIP, a VQA framework built entirely upon CVLMs. A Shared Cross-Modal Adapter (SCMA) is employed to optimize feature representations in both the visual and textual branches. Thanks to its minimal number of trainable parameters, SCMA significantly reduces the training cost. Additionally, a learnable five-level prompt mechanism is introduced to help the model perceive fine-grained quality variations. Furthermore, we investigate the impact of different frame sampling strategies on VQA. Experimental results demonstrate that Q-CLIP outperforms existing methods on multiple VQA datasets.

## Acknowledgements

This work was supported by the National Natural Science Foundation of China under Grant 62441202, the Natural Science Foundation of Jiangsu under Grant BK20250731, the General Program of China Postdoctoral Science Foundation under grant 2025M784450, the Startup Foundation for Introducing Talent of Nanjing University of Information Science and Technology under grant 2025r029.

## Impact Statement

This paper presents work whose goal is to advance the field of Machine Learning. There are many potential societal consequences of our work, none which we feel must be specifically highlighted here.

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

## A. Methods Comparison

To clarify the novelty of our approach, we visually compare our method with existing VQA methods that utilize CVLMs in Fig. 7. In existing VQA methods leveraging CVLMs (as shown in Fig. 7(b)), CVLMs serve as an auxiliary component. They provide supplementary features to enhance the performance of the backbone network. The final quality score is derived through a process of feature aggregation that combines information from both the backbone network and the CVLMs. In contrast, our method (as shown in Fig. 7(a)), which is the first VQA approach fully based on CVLMs, directly employs CVLMs to process the input and generate the quality score without relying on any additional backbone network or feature aggregation with other network components. This fundamental difference highlights the originality of our work in solely harnessing the capabilities of CVLMs for VQA tasks.

**CVLMs vs. MLLMs.** With the rapid progress of MLLMs/LMMs, recent works (Zhang et al., 2024b; 2025; Wu et al., 2024) have leveraged these *generative* models for image/video quality assessment (IQA/VQA) by prompting them to autoregressively produce quality-related text (e.g., descriptions/rationales or text-defined rating levels). In contrast, our method is built on *contrastive* vision-language models (CVLMs), i.e., CLIP-style dual encoders, and performs *discriminative, similarity-based* prediction in a shared embedding space rather than free-form decoding; hence it does not rely on autoregressive generation. Since some recent literature loosely uses the term "VLM" to refer to both CLIP-style encoders and MLMMs-based generative models, we consistently use *CVLMs* to avoid ambiguity and to distinguish our backbone and inference mechanism from MLLMs-based approaches.

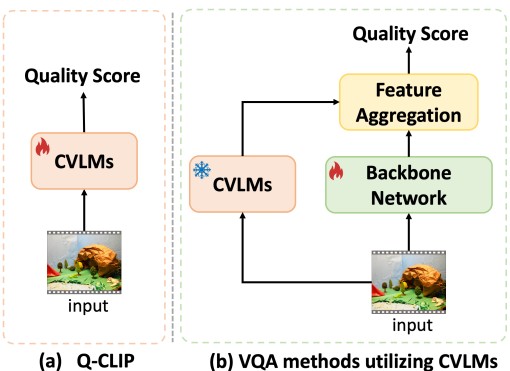

*Figure 7.* Methods comparison.

## B. Efficiency

We conduct a comprehensive analysis of the model's runtime efficiency, as shown in Tab. 7. Specifically, we compare the FLOPs, GPU runtimes, and throughput for videos of different resolutions, where the length of the videos is 150 frames. Compared to CNN-based models (VSFA, PVQ, BVQA), Q-CLIP reduces FLOPs by up to **6.6×**, **9.5×**, and **18.3×**, as well as reduces computation time by up to **17.1×**, **21.3×**, and **46.4×**, respectively. Furthermore, compared to methods utilizing CVLMs like CLiF-VQA, MaxVQA, and CLIPVQA, our method offers significantly improved processing efficiency, being **2.7×** faster than the state-of-the-art CLIPVQA.

*Table 7.* FLOPs, running time and throughput (average of 10 runs) on RTX 4090. FLOPs, running time and throughput are in $G$, $s$ and $videos/s$, respectively.

| Methods | 540p | | | 720p | | | 1080p | | |
|---|---|---|---|---|---|---|---|---|---|
| | FLOPs | Time | Throughput | FLOPs | Time | Throughput | FLOPs | Time | Throughput |
| VSFA | 6440 | 0.672 | 1.488 | 11426 | 1.141 | 0.876 | 25712 | 2.362 | 0.423 |
| PVQ | 9203 | 0.812 | 1.232 | 13842 | 1.334 | 0.750 | 36760 | 2.935 | 0.341 |
| BVQA | 17705 | 1.414 | 0.707 | 31533 | 3.579 | 0.279 | 70714 | 6.402 | 0.156 |
| FAST-VQA | 284 | 0.025 | 40.00 | 284 | 0.025 | 40.00 | 284 | 0.025 | 40.00 |
| DOVER | 282 | 0.031 | 32.26 | 282 | 0.031 | 32.26 | 282 | 0.031 | 32.26 |
| Q-Align | 6242 | 0.753 | 1.328 | 6242 | 0.755 | 1.325 | 6242 | 0.752 | 1.330 |
| MBVQA | 912 | 0.212 | 4.717 | 1232 | 0.249 | 4.016 | 2150 | 0.353 | 2.833 |
| CLiF-VQA | 1432 | 0.233 | 4.292 | 1432 | 0.233 | 4.292 | 1432 | 0.233 | 4.292 |
| MaxVQA | 693 | 0.161 | 6.211 | 693 | 0.161 | 6.211 | 693 | 0.161 | 6.211 |
| CLIPVQA | 3845 | 0.377 | 2.653 | 3845 | 0.376 | 2.660 | 3845 | 0.376 | 2.660 |
| **Q-CLIP** | 3870 | 0.138 | 7.246 | 3870 | 0.138 | 7.246 | 3870 | 0.138 | 7.246 |

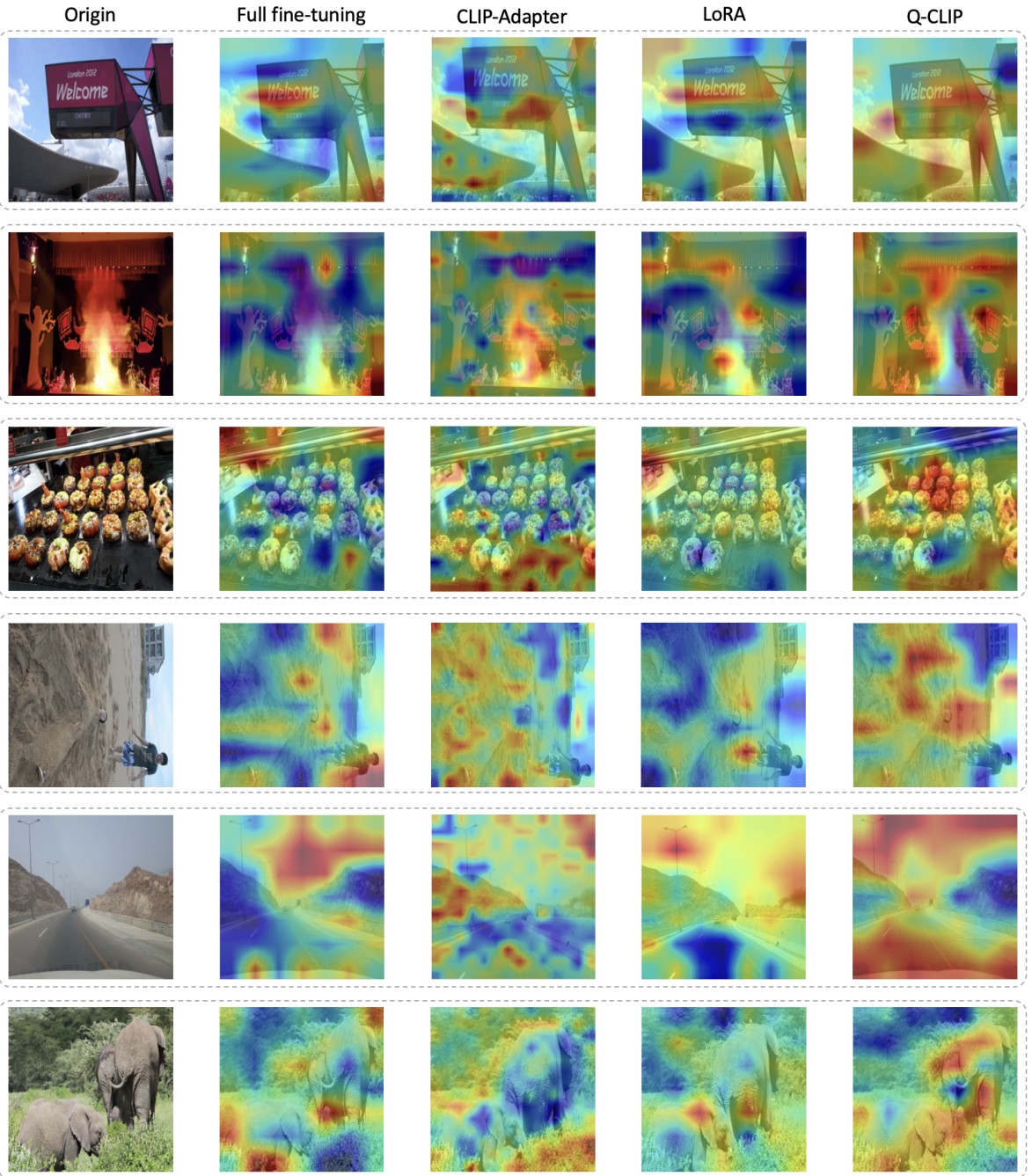

*Figure 8.* Comparison of attention visualizations.

## C. More Visualizations

We compare the attention visualizations of Q-CLIP with full fine-tuning, CLIP-Adapter, and LoRA to illustrate how Q-CLIP attends to quality-relevant regions, as shown in Fig. 8. In the visualization process, we use video frames as input to the visual encoder and provide a guiding textual prompt (such as "a video of high quality") as input to the text encoder. Each video frame is divided into multiple visual patches, and a feature vector is extracted for each patch, resulting in a patch-level visual representation of the frame. At the same time, a global semantic embedding is obtained from the text input. We then compute the similarity between each patch feature and the text embedding, where this similarity score reflects the semantic relevance between the visual patch and the text prompt. A higher score indicates a stronger semantic association as perceived by the model. Based on these similarity scores, we construct an attention map by mapping each patch's score to a visually distinguishable color: typically, warmer colors (such as red or yellow) indicate higher attention weights, while cooler colors (such as blue or green) indicate lower ones. This results in an attention heatmap that visualizes the model's focus with respect to the given textual guidance.

From the visualization, it is evident that the attention distributions of full fine-tuning, CLIP-Adapter, and LoRA fail to align well with the attention requirements for quality perception. Full fine-tuning either scatters attention across non-critical areas or focuses on semantic regions irrelevant to quality assessment. While CLIP-Adapter and LoRA exhibit greater attention to non-semantic regions compared to full fine-tuning, they remain deficient in capturing quality-relevant regions. This is particularly notable given that such non-semantic regions inherently possess quality-relevant characteristics, yet both methods fail to adequately capture the comprehensive quality information across the entire image. In contrast, Q-CLIP not only attends to semantic locations but also highlights multiple critical areas within video frames, resulting in a more concentrated and quality-relevant attention distribution. This suggests that, with the integration of its cross-modal adapters and other design elements, Q-CLIP achieves a more precise understanding of quality-related semantics and better captures their visual correlations. Consequently, its attention guidance for quality perception in video quality assessment is more reasonable and effective. This indicates that the model's ability to focus on critical quality-related features aligns better with the inherent characteristics of video quality evaluation, thereby enhancing both the rationality and efficacy of the assessment process.

## D. More Experimental Details

### D.1. Additional Model Details

Specifically, we add 6 layers of SCMA to the model. In the ablation study on the number of layers, the counting is done in reverse order. That is, the first SCMA layer corresponds to the last encoder layer, while the sixth SCMA layer corresponds to the sixth-to-last encoder layer. The FFN hidden dimension in both E-SCMA and P-SCMA is set to 128.

### D.2. Details of Fine-tuning

To achieve better performance on small datasets, we adopt different fine-tuning strategies for each dataset. Specifically, for LIVE-VQC, we fine-tune only the P-SCMA module with a learning rate of 0.002. For KoNViD-1k, we train only the FFN component of E-SCMA with a learning rate of 0.001. For CVD2014 and LIVE-Qualcomm, we fine-tune only the down and up projection layers of E-SCMA, both with a learning rate of 0.001. For YouTube-UGC, we fine-tune the entire SCMA module, using separate learning rates for different components: 0.0001 for E-SCMA and 0.001 for P-SCMA. All experiments are conducted with a batch size of 12 for 20 epochs. The learning rate is scheduled using cosine annealing, with a 4-epoch warm-up phase at the beginning.

### D.3. Single Unified Fine-Tuning Strategy Already Achieves SOTA

In the main paper, we adopt slightly different fine-tuning hyperparameters for each small dataset (LIVE-VQC, KoNViD-1k, CVD2014, YouTube-UGC, LIVE-Qualcomm) in order to obtain a small additional performance gain. To verify that our method does not rely on such dataset-specific tuning, we further conduct experiments with a single unified fine-tuning strategy for all these datasets. Specifically, we fine-tune the entire 0.14M-parameter SCMA module using the same learning rate and training schedule on every small dataset, without any dataset-specific choices. The results in Tab. 8 show that, under this unified setting, Q-CLIP still achieves state-of-the-art performance on all small datasets and remains clearly stronger than prior VQA methods. This confirms that Q-CLIP is robust to the choice of fine-tuning strategy and that the per-dataset strategies in the main paper should be viewed as optional practical refinements rather than a requirement.

*Table 8.* Comparison of fine-tuning strategies on small datasets.

| Datasets | LIVE-VQC | | KoNViD-1k | | YouTube-UGC | | CVD2014 | | LIVE-Qualcomm | |
|---|---|---|---|---|---|---|---|---|---|---|
| Methods | SRCC↑ | PLCC↑ | SRCC↑ | PLCC↑ | SRCC↑ | PLCC↑ | SRCC↑ | PLCC↑ | SRCC↑ | PLCC↑ |
| SCMA (Unified) | 0.879 | 0.900 | 0.913 | **0.920** | **0.911** | **0.911** | **0.898** | 0.906 | 0.844 | 0.881 |
| Q-CLIP (Per-dataset) | **0.881** | **0.901** | **0.915** | **0.920** | **0.911** | **0.911** | 0.897 | **0.907** | **0.846** | **0.884** |

## D.4. Loss Function

In line with mainstream VQA methods (Wu et al., 2022; 2023a; Mi et al., 2024b; Zhao et al., 2023; Liu et al., 2024; Mi et al., 2024a), we utilize both monotonicity- and linearity-driven loss components, which are widely recognized as essential for quality prediction. The loss function used to optimize the proposed models consists of two parts: the monotonicity-induced loss and linearity-induced loss. Given m predicted quality scores $\hat{Q} = \{\hat{q_1}, \hat{q_2}, ..., \hat{q_m}\}$ and m ground-truth subjective quality scores $Q = \{q_1, q_2, ..., q_m\}$.

Specifically, the monotonicity-induced loss predicts the monotonicity of the video quality scores by introducing additional order constraints. The monotonicity-induced loss function is defined as follows:

$$L_{mon} = \frac{1}{m^2} \sum_{i=1}^{m} \sum_{j=1}^{m} max(0, |q_i - q_j| - f(q_i, q_j) \cdot (\hat{q}_i - \hat{q}_j)) \tag{14}$$

where $f(q_i, q_j) = 1$ if $q_i \geq q_j$, otherwise $f(q_i, q_j) = -1$.

In contrast, the goal of the linearity-induced loss is to compute the linear relationship between the predicted quality score and ground-truth subjective quality score. The linearity-induced loss function can be denoted as:

$$L_{lin} = (1 - \frac{\sum_{i=1}^{m}(\hat{q}_i - \hat{a})(q_i - a)}{\sqrt{\sum_{i=1}^{m}(\hat{q}_i - \hat{a})^2 \sum_{i=1}^{m}(q_i - a)^2}})/2 \tag{15}$$

where $a = \frac{1}{m}\sum_{i=1}^{m} q_i$ and $\hat{a} = \frac{1}{m}\sum_{i=1}^{m} \hat{q}_i$.

Finally, the total loss function $L$ is obtained by combining the two loss functions $L_{mon}$ and $L_{lin}$ above:

$$L = \alpha L_{mon} + \beta L_{lin} \tag{16}$$

where $\alpha = 0.3$ and $\beta = 1$ represent the weights of monotonicity-induced loss and linearity-induced loss.

## E. Explanation of Sampling

To provide a clearer understanding of the various sampling strategies explored in this work, we describe them in detail as follows:

**Uniform Sampling.** The video is evenly divided into 8 segments, and the middle frame of each segment is selected.

**Random Sampling.** The video is evenly divided into 8 segments, and one frame is randomly selected from each segment.

**Uniform Sampling with Random Start.** The video is evenly divided into 8 segments. A random frame is selected from the first segment, and the corresponding frame at the same relative position is selected from the remaining segments.

**Equal Intervals + Per-Segment MSE Average.** The video is evenly divided into 8 segments. From each segment, the frame whose MSE is closest to the average MSE of all frames in that segment is selected.

**Equal Intervals + Per-Segment MSE Median.** The video is evenly divided into 8 segments. From each segment, the frame with the median MSE value is selected.

**Uniform sampling over MSE-ranked frames.** All frames are ranked based on their MSE values. The ranked list is divided into 8 equal parts, and the middle frame from each part is selected.

**Definition of "Q-CLIP-Mixed" Sampling Strategy.** Q-CLIP-Mixed is a mixture of all six frame sampling strategies. It is used in both training and inference as follows:

- **Training.** For each video in each iteration, we randomly select one of the six sampling strategies with equal probability (i.e., a uniform 1/6 chance for each strategy) to extract frame subsets. In this way, the model is exposed to diverse sampling patterns and learns to be robust to how frames are selected, rather than overfitting to a single fixed strategy.

- **Inference.** We simultaneously apply all six sampling strategies to the video, then use the average of all sampled predictions as the final prediction. Therefore, Q-CLIP-Mixed can be regarded as a simple ensemble method that combines the six sampling schemes with equal weights.

## F. Additional Ablation Studies

### F.1. Ablation on P-SCMA

We validate the effectiveness of integrating P-SCMA into the projection component, as shown in Tab. 9. The results show that incorporating P-SCMA into the model significantly enhances performance. In addition, we conduct an ablation study on the number of FFN layers in P-SCMA, as shown in Tab. 10. A simple structure that only reduces and then expands the feature dimension yields suboptimal performance. Introducing a single intermediate mapping layer leads to a notable performance improvement. However, increasing the number of layers beyond one does not bring further benefits. In fact, performance begins to decline when the depth reaches three layers. This is likely because the projection is designed for a straightforward transformation to support similarity computation, and adding too many parameters may cause overfitting.

*Table 9.* Ablation on P-SCMA.

| Datasets | $\text{LSVQ}_{test}$ | | $\text{LSVQ}_{1080p}$ | | KoNViD-1k | |
|---|---|---|---|---|---|---|
| P-SCMA | SRCC↑ | PLCC↑ | SRCC↑ | PLCC↑ | SRCC↑ | PLCC↑ |
| *w/o* | 0.886 | 0.885 | 0.815 | 0.844 | 0.877 | 0.882 |
| *w/* | **0.897** | **0.895** | **0.820** | **0.853** | **0.883** | **0.891** |

*Table 10.* Ablation on the number of FFN layers in P-SCMA.

| Datasets | $\text{LSVQ}_{test}$ | | $\text{LSVQ}_{1080p}$ | | KoNViD-1k | |
|---|---|---|---|---|---|---|
| layers | SRCC↑ | PLCC↑ | SRCC↑ | PLCC↑ | SRCC↑ | PLCC↑ |
| *0* | 0.891 | 0.889 | 0.816 | 0.848 | 0.879 | 0.886 |
| *1* | **0.897** | **0.895** | **0.820** | **0.853** | **0.883** | **0.891** |
| *2* | **0.897** | **0.895** | **0.820** | 0.852 | 0.882 | 0.890 |
| *3* | 0.895 | 0.893 | 0.818 | **0.853** | 0.880 | 0.888 |
| *4* | 0.894 | 0.891 | 0.818 | 0.852 | 0.878 | 0.886 |

### F.2. Ablation on FFN Dimension in SCMA

We perform an ablation study on different FFN dimensions in SCMA to assess their effect on model performance. The results are shown in Tab. 11. When the FFN dimension is set too low (below 64), the model performs poorly. As the dimension increases, the performance gradually improves, indicating that too small a dimension is insufficient for learning adequate feature representations. The best performance is achieved at a dimension of 128. However, further increasing the dimension leads to a decline in performance, suggesting that excessively high dimensions increase the number of learnable parameters and cause the model to overfit.

*Table 11.* Ablation on FFN dimension in SCMA.

| Datasets | $\text{LSVQ}_{test}$ | | $\text{LSVQ}_{1080p}$ | | KoNViD-1k | |
|---|---|---|---|---|---|---|
| Dim | SRCC↑ | PLCC↑ | SRCC↑ | PLCC↑ | SRCC↑ | PLCC↑ |
| *16* | 0.889 | 0.888 | 0.812 | 0.843 | 0.878 | 0.887 |
| *32* | 0.894 | 0.891 | 0.815 | 0.844 | 0.880 | 0.889 |
| *64* | 0.896 | **0.895** | **0.820** | 0.850 | **0.883** | 0.890 |
| *128* | **0.897** | **0.895** | **0.820** | **0.853** | **0.883** | **0.891** |
| *256* | 0.896 | **0.895** | **0.820** | 0.852 | 0.882 | **0.891** |
| *512* | 0.893 | 0.891 | 0.818 | 0.851 | 0.879 | 0.888 |

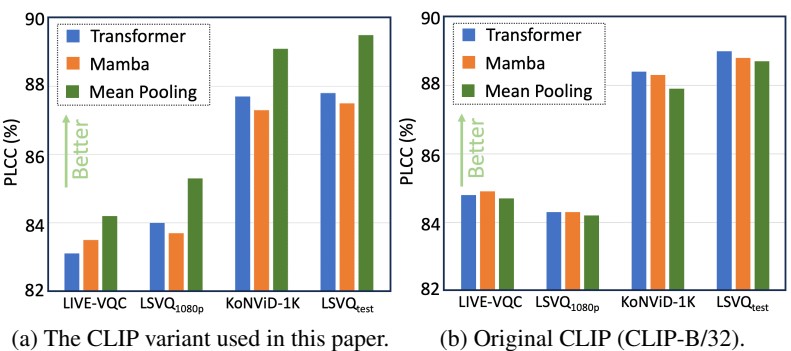

*Figure 9.* Ablation on Frame Feature Fusion.

(a) The CLIP variant used in this paper.   (b) Original CLIP (CLIP-B/32).

*Figure 10.* Comparison of different fusion strategies on LSVQ.

## F.3. Ablation on Temporal Aggregation of Frame Features

To study how temporal aggregation affects Q-CLIP, we compare three representative strategies for fusing frame-level features: (i) a Transformer-based temporal head (Vaswani et al., 2017), (ii) a Mamba-based temporal head (Gu & Dao, 2023), and (iii) zero-parameter mean pooling, as shown in Fig. 9. We evaluate each strategy under two backbones: a video-tuned CLIP variant used by Q-CLIP and the original CLIP (CLIP-B/32). Results are summarized in Fig. 10 for four datasets (LIVE-VQC, $LSVQ_{1080p}$, KoNViD-1K, and $LSVQ_{test}$), reported in terms of PLCC.

Fig. 10(a) shows that when using the *video-tuned* backbone, mean pooling consistently yields the best performance across all evaluated datasets, outperforming both Transformer and Mamba temporal heads by a clear margin. In contrast, Fig. 10(b) indicates that under the *original* CLIP backbone, the performance gaps among aggregation strategies become smaller and can be dataset-dependent (Transformer/Mamba may match or slightly exceed pooling on some benchmarks), while mean pooling remains highly competitive overall. These results suggest that the effectiveness of temporal aggregation is tightly coupled with the backbone's pre-training objective and representation space. Our primary backbone is *pre-finetuned on large-scale video data using mean-pooled frame embeddings*. In this setting, per-frame representations already encode motion patterns and temporal consistency in a way that is compatible with mean pooling. Therefore, retaining mean pooling preserves the inductive bias and geometry of the learned representation space, whereas inserting an additional trainable temporal module (Transformer/Mamba) may alter the feature distribution and disrupt the pretrained structure, leading to suboptimal transfer (as shown in Fig. 10(a)).

Although mean pooling is a simple aggregator, Q-CLIP still exploits temporal cues through two complementary mechanisms. First, Q-CLIP operates on multiple frames rather than a single frame, and our frame-difference-based sampling explicitly selects frames that best represent motion changes, injecting temporal information before aggregation. Second, the video-tuned backbone has already learned to embed temporal dynamics into frame-level features under a mean-pooling objective; thus, a lightweight aggregator can be sufficient for capturing temporally-related distortions when coupled with motion-aware sampling and a video-pretrained encoder.

From a parameter-efficiency perspective, mean pooling introduces no extra trainable parameters, which is particularly favorable under limited training data. As also observed in prior CVLMs-based video studies (e.g., CLIP4CLIP (Luo et al., 2022)), simple pooling can outperform heavier temporal models on small-scale benchmarks, suggesting that adding temporal parameters may be unnecessary or even harmful due to overfitting. This aligns with our goal of keeping Q-CLIP extremely lightweight to train while maintaining strong generalization across datasets.

Based on the above evidence, we adopt mean pooling as the default temporal aggregation strategy in Q-CLIP. It is not only the most effective choice for our video-tuned backbone (Fig. 10(a)), but also a robust and regularized option that avoids unnecessary temporal parameters and mitigates overfitting risk.

## F.4. Limited-Data Robustness

Evaluating performance under limited data is crucial for VQA methods. Thus, we have added extended experiments, where we randomly sub-sample the LSVQ training set to 20%, 50%, and 80%, keeping all other training settings identical to the full-data case. It is important to note that we do not employ mixed sampling to enhance model performance, but rather use the most basic uniform sampling. Detailed results are shown in Tab. 12. As can be seen, when using only 20% of the training data, Q-CLIP has achieved performance comparable to or even better than strong baselines. As the training ratio increases from 20% → 50% → 80% → 100%, the performance improves smoothly, with only marginal gains beyond 50%, indicating that Q-CLIP quickly saturates and uses labelled data efficiently. In particular, when using 80% of the training data, Q-CLIP achieves performance comparable to that obtained with 100% of the training data. These observations demonstrate that Q-CLIP is indeed robust and data-efficient under limited supervision.

*Table 12.* Ablation on limited-data.

| Datasets | $\text{LSVQ}_{test}$ | | $\text{LSVQ}_{1080p}$ | | KoNViD-1k | | LIVE-VQC | |
|---|---|---|---|---|---|---|---|---|
| Data Ratio | SRCC↑ | PLCC↑ | SRCC↑ | PLCC↑ | SRCC↑ | PLCC↑ | SRCC↑ | PLCC↑ |
| **Q-CLIP-20%** | 0.878 | 0.881 | 0.798 | 0.837 | 0.869 | 0.876 | 0.804 | 0.834 |
| **Q-CLIP-50%** | 0.888 | 0.890 | 0.817 | 0.850 | 0.882 | 0.889 | 0.815 | 0.844 |
| **Q-CLIP-80%** | 0.894 | **0.895** | 0.816 | 0.850 | **0.890** | **0.896** | **0.816** | **0.849** |
| **Q-CLIP-100%** | **0.897** | **0.895** | **0.820** | **0.853** | 0.883 | 0.891 | 0.803 | 0.842 |

## F.5. Generalizability of Prompts and Sampling

To better isolate the core contributions of the Q-CLIP framework, we test the proposed prompt and sampling strategies on other architectures.

### (1) Learnable Five-Level Prompts on a CVLMs-Utilizing Baseline (MaxVQA).

We take MaxVQA (Wu et al., 2023c) as a representative CLIP-utilizing VQA model and replace its original antonym-style text prompts with our learnable five-level prompts ("excellent/good/fair/poor/bad"), while keeping all other settings the same as in the MaxVQA paper. Under this setup, we re-train MaxVQA on LIVE-VQC, KoNViD-1k, and YouTube-UGC. As shown in Tab. 13. Across all three datasets, the introduction of our prompting strategy brings consistent performance improvements over the original MaxVQA. This confirms that the proposed prompt design is a generally valuable component that can benefit other CLIP-utilizing VQA models, not just Q-CLIP.

*Table 13.* Learnable five-level prompts on MaxVQA.

| Datasets | LIVE-VQC | | KoNViD-1k | | YouTube-UGC | |
|---|---|---|---|---|---|---|
| Methods | SRCC↑ | PLCC↑ | SRCC↑ | PLCC↑ | SRCC↑ | PLCC↑ |
| MaxVQA | 0.854 | 0.873 | 0.894 | 0.895 | 0.894 | 0.890 |
| **MaxVQA + Our Prompt Strategy** | **0.862** | **0.879** | **0.903** | **0.905** | **0.907** | **0.902** |

### (2) Frame Sampling on the Most Classical Baseline (FAST-VQA).

We further apply our mixed frame sampling strategy to FAST-VQA (Wu et al., 2022), while keeping its original spatial sampling scheme unchanged. Again, using the official training settings of FAST-VQA, we evaluate on LIVE-VQC, KoNViD-1k, and YouTube-UGC. As shown in Tab. 14. The modified FAST-VQA with our sampling strategy consistently outperforms the original FAST-VQA across all three benchmarks, indicating that our sampling design is also broadly helpful for conventional video-backbone-based VQA methods.

*Table 14.* Frame sampling on FAST-VQA.

| Datasets | LIVE-VQC | | KoNViD-1k | | YouTube-UGC | |
|---|---|---|---|---|---|---|
| Methods | SRCC↑ | PLCC↑ | SRCC↑ | PLCC↑ | SRCC↑ | PLCC↑ |
| FAST-VQA | 0.845 | 0.852 | 0.890 | 0.889 | 0.857 | 0.853 |
| **FAST-VQA + Our Sampling Strateg** | **0.860** | **0.862** | **0.899** | **0.900** | **0.866** | **0.868** |

## G. Theoretical Perspectives of SCMA

From the representation learning perspective, we can view SCMA as a shared residual transformation of the joint space in CVLMs.

Let $f_v(x)$ and $f_t(y)$ denote the frozen visual and textual embeddings from the CVLMs backbone. SCMA applies the same residual adapter $A(\cdot)$ to both modalities:

$$z_v = f_v(x) + A\big(f_v(x)\big), z_t = f_t(y) + A\big(f_t(y)\big) \tag{17}$$

The quality-related similarity used by Q-CLIP is then $\langle z_v, z_t \rangle$. If we locally linearize the adapter (a standard first-order approximation), we may view $A(f) \approx Jf + b$ around the current feature region, leading to a shared linear transform $T = I + J$ and

$$\langle z_v, z_t \rangle \approx \langle Tf_v, Tf_t \rangle = f_v^\top (T^\top T) f_t \tag{18}$$

Thus, in this local view, SCMA effectively applies a shared linear transform $T$ to the frozen CLIP embeddings, inducing a Mahalanobis-style metric $M = T^\top T$ in the joint space. Training SCMA with MOS supervision can therefore be interpreted as learning a quality-aware metric on top of CLIP, rather than re-learning the entire encoder. Because the same adapter is shared between visual and textual branches, both modalities are forced to live in the same transformed coordinate system, which explicitly reduces the visual–textual gap for quality. The five quality-level prompts (excellent/good/fair/poor/bad) then act as prototypes in this transformed space, and SCMA learns to map video embeddings toward the appropriate quality region, strengthening the model's ability to distinguish fine-grained quality differences rather than only semantic differences.

## H. Additional Experiments on Recent Datasets

To further examine the robustness of Q-CLIP, we conduct additional evaluations on two recently released VQA datasets that we were able to obtain at the time of submission: **KVQ** (Zhou et al., 2022c) and **YouTube SFV+HDR** (Wang et al., 2024). We compare Q-CLIP against the best-performing methods reported in the literature for these datasets, including FAST-VQA, DOVER and KSVQE, as shown in Tab. 15.

*Table 15.* Additional results on KVQ and YouTube SFV+HDR.

| Datasets | KVQ | | YouTube SFV+HDR | |
|---|---|---|---|---|
| Methods | SRCC↑ | PLCC↑ | SRCC↑ | PLCC↑ |
| FAST-VQA | 0.832 | 0.834 | 0.752 | 0.797 |
| DOVER | 0.833 | 0.837 | 0.702 | 0.781 |
| KSVQE | 0.867 | 0.869 | – | – |
| **Q-CLIP** | **0.877** | **0.878** | **0.804** | **0.825** |

Q-CLIP achieves the best performance on both datasets, suggesting strong generalization beyond the benchmarks considered in the main paper. These additional results further support the robustness of our approach across diverse content types and distortion patterns.

