# OpenReview forum: "Q-CLIP: Unleashing the Power of Vision-Language Models for Video Quality Assessment through Unified Cross-Modal Adaptation"
_ICML.cc/2026/Conference — ICML 2026 regular_

### Official Review · Reviewer_qBVx · 2026-02-25

**Soundness:** 2
**Presentation:** 3
**Significance:** 3
**Originality:** 2
**Overall Recommendation:** 4
**Confidence:** 3

**Summary:**

This paper introduces Q-CLIP, a CVLM-based framework for VQA. Q-CLIP adapts the entire framework using a Shared Cross-Modal Adapter. The architecture incorporates three key design elements: an encoder-side adapter, a learnable five-level prompt mechanism, and an exploration of frame sampling strategies. Q-CLIP achieves state-of-the-art performance on multiple VQA benchmarks while maintaining high parameter efficiency.

**Compliance With Llm Reviewing Policy:**

Affirmed.

**Final Justification:**

All of my concerns are well addressed. I choose to increase the score by 1.

**Key Questions For Authors:**

1. Beyond parameter efficiency, what are the specific representational advantages of the "shared" mechanism in SCMA compared to a simple projector or modality-specific adapters?
2. Why does the theoretical analysis in Appendix G treat SCMA as a linear transform when the architecture includes nonlinear GELU layers?
3. Can the SCMA design be generalized to other fine-grained cross-modal tasks, or does it rely specifically on VQA domain knowledge?

**Limitations:**

yes

**Strengths And Weaknesses:**

**Strengths**
1. The method is extensively validated across various benchmarks, and the effectiveness of its core modules is supported by detailed ablation studies.
2. Q-CLIP is highly efficient, training only a minimal number of parameters compared to leading VQA models.
3. The paper is well-structured, and the proposed modules are straightforward and easy to implement.

**Weakness**
1. The concepts of five-level quality scales and motion-based frame sampling are relatively established in VQA and MLLM-based quality research. The paper could more clearly articulate the novel insights or unique combinations of these strategies within the SCMA context.
2. While SCMA is shown to be effective, the paper lacks a deep investigation into why shared cross-modal adaptation is superior to separate modality-specific adapters or a simple projection layer.
3. The theoretical explanation in Appendix G reportedly treats SCMA as a linear transform $T$. However, the actual implementation uses nonlinear components like GELU. This discrepancy between the theoretical simplification and the practical architecture should be addressed to provide a better analysis for SCMA.

---

> ### Author Rebuttal · Authors · 2026-03-29
>
> ### **We sincerely thank the reviewer for the careful and constructive comments, and positive recognition of our empirical validation, parameter efficiency, and paper organization.**
>
> ### **Question 1: Unique role of five-level quality scales and motion-based frame sampling. (W1)**
>
> ### **Answer：**
> Thank you for this insightful comment. We would like to clarify the roles of these two components in our paper. Motion-based frame sampling is not claimed as a core methodological novelty. In this work, **we study different sampling strategies mainly as a systematic empirical analysis to understand how sampling affects VQA performance and cross-dataset generalization. Our goal here is to provide practical evidence on sampling behavior, rather than to present sampling itself as a main contribution.** In contrast, **the main novelty of our method lies in the joint design of SCMA and learnable five-level prompts within a fully CVLM-based VQA framework.** The five-level quality scales are not used as a standalone idea, but are introduced as learnable quality prompts that work together with SCMA for fine-grained quality modeling. Specifically, SCMA adapts both the visual and textual branches, while the five-level prompts provide ordered quality-aware text representations. Their similarities with video features are then organized into a structured quality representation for continuous MOS prediction. Therefore, the contribution is not simply the use of five quality levels, but their integration with shared cross-modal adaptation to better model subtle quality differences in VQA.
>
> ### **Question 2: Advantages of the shared mechanism over a simple projector or modality-specific adapters. (W2 & Q1)**
>
> ### **Answer：**
> Thank you for this important comment. From a mechanistic perspective, **the key role of the shared design in SCMA is to reduce the modality gap and improve cross-modal consistency.** Prior studies [1-3] have shown that **the performance of CLIP-like models in downstream tasks is often limited by the gap between visual and textual representations, and reducing this gap is crucial for improving transfer performance.**
>
> **Modality-specific adapters may drive the two branches toward different modality-dependent directions, while a simple projector only changes the final mapping and cannot correct the mismatch on the encoder side. In contrast, SCMA applies a shared adaptation rule to both branches, so the two modalities are adjusted more consistently before similarity computation, which is better suited to fine-grained VQA.**
>
> This is consistent with our design and evidence: as shown in Table 4, SCMA demonstrates significantly better performance than the non-shared dual-branch adaptation scheme; it outperforms non-shared dual-branch adaptation as well as CLIP-Adapter and LoRA. Figure 6 further shows a clearer visual-text structure after adaptation.
>
> [1] Mind the gap Understanding the modality gap in multi-modal contrastive representation learning. NeurIPS 2022.
>
> [2] Intra-modal proxy learning for zero-shot visual categorization with clip. NeurIPS 2023.
>
> [3] Post-pre-training for modality alignment in vision-language foundation models. CVPR 2025.
>
>
> ### **Question 3: Linear transform in Appendix G despite nonlinear GELU layers. (W3 & Q2)**
>
> ### **Answer：**
>
> We appreciate the reviewer for this careful comment. **The key clarification is that Appendix G treats SCMA as a linear transform only as a local first-order view, rather than as an exact form of the implemented module**. This is already stated in the appendix by “*If we locally linearize the adapter (a standard first-order approximation)*,” after which the shared residual adapter is written as a local transform $T = I + J$. Under this view, the linear transform $T$ is introduced to explain the main shared residual effect of SCMA in the joint visual-text space, namely, how shared adaptation induces a common transformed space or metric for quality prediction. It is therefore an interpretive approximation used for analysis, not a claim that the full SCMA is strictly linear. This is also consistent with the main formulation, where the shared core is defined as a general $f_\theta$. We will revise the wording to make this point explicit.
>
>
> ### **Question 4: Generalizability of SCMA beyond VQA. (Q3)**
>
> ### **Answer：**
> Thank you for this insightful comment. **SCMA itself does not rely on VQA-specific domain knowledge.** It is a shared representation-adaptation module defined on top of a frozen CVLM, and its role is to jointly refine visual and textual features without introducing task-specific modality rules. In our framework, the VQA-specific parts are mainly the five-level prompts and the final MOS regression, rather than the shared adapter principle itself. Therefore, **SCMA is general in design and can in principle be applied to other fine-grained cross-modal tasks where subtle visual-text alignment is important.**

---

> > ### Author Rebuttal · Reviewer_qBVx · 2026-04-03
> >
> > Can you provide some empirical result on the generalizability of SCMA?

---

> > > ### Author Response · Authors · 2026-04-04
> > >
> > > To provide direct evidence, we evaluated the same SCMA design beyond the current VQA setting on three additional fine-grained visual assessment tasks: Image Quality Assessment (IQA), Point Cloud Quality Assessment (PCQA), and Image Aesthetic Assessment (IAA). As shown in Tables 1–3, the results are reported in terms of SROCC/PLCC. Notably, our method achieves SOTA performance on all three tasks under their respective standard benchmarks and evaluation protocols.
> > >
> > > These results suggest that the benefit of SCMA is not tied to the specific formulation of our VQA framework. Instead, SCMA provides a generally effective way to model fine-grained cross-modal interactions through a shared adaptation space, which transfers well across diverse visual assessment tasks. Therefore, we believe that these additional results provide direct empirical support for the generalizability of SCMA and substantially strengthen the claim that SCMA is a transferable cross-modal adaptation mechanism rather than a task-specific design choice.
> > >
> > >
> > > **Table 1. Experimental performance on IQA.**
> > > | Methods           |     LIVE    |   KADID-10k |    LIVEC    |  KonIQ-10k  |
> > > |:-----------------:|:-----------:|:-----------:|:-----------:|:-----------:|
> > > | LoDa (CVPR2024)   | 0.975/0.979 | 0.931/0.936 | 0.876/0.899 | 0.932/0.944 |
> > > | PCIQA (TMM2024)   | 0.976/0.976 | 0.870/0.866 | 0.863/0.880 | 0.918/0.930 |
> > > | QAL-IQA (IJCV2025)| 0.971/0.973 | 0.908/0.910 | 0.859/0.875 | 0.917/0.928 |
> > > | **Q-CLIP**        |**0.983/0.983**|**0.948/0.950**|**0.907/0.925**|**0.942/0.953**|
> > >
> > >
> > >
> > > **Table 2. Experimental performance on PCQA.**
> > > |        Methods         |   SJTU-PCQA   |    LS-PCQA    |
> > > |:----------------------:|:-------------:|:-------------:|
> > > |    3DTA (TMM2024)      |  0.931/0.953  |  0.604/0.613  |
> > > |  AFQ-Net (TCSVT2025)   |  0.930/0.957  |  0.680/0.690  |
> > > | CLIP-PCQA (AAAI2025)   |  0.936/0.956  |  0.736/0.755  |
> > > |      **Q-CLIP**        |**0.958/0.963**|**0.756/0.772**|
> > >
> > >
> > >
> > > **Table 3. Experimental performance on IAA.**
> > > |        Methods         |     AVA      |
> > > |:----------------------:|:------------:|
> > > |   VILA-R (CVPR2023)   |  0.774/0.774  |
> > > |     SAGAN (TMM2024)     |  0.774/0.788  |
> > > |     Charm (CVPR2025)    |  0.781/0.783  |
> > > |      **Q-CLIP**         | **0.810/0.831**|

---

### Official Review · Reviewer_Pxyv · 2026-03-11

**Soundness:** 3
**Presentation:** 3
**Significance:** 3
**Originality:** 3
**Overall Recommendation:** 5
**Confidence:** 4

**Summary:**

This paper proposes Q-CLIP, a video quality assessment framework that adapts pretrained contrastive vision-language models to the VQA task through a lightweight Shared Cross-Modal Adapter. The adapter is inserted into both visual and textual branches while keeping the backbone frozen, enabling parameter-efficient fine-tuning with only a small number of trainable parameters. The method further introduces five learnable quality prompts to guide the model in capturing fine-grained perceptual quality via cross-modal similarity. Experiments across six VQA datasets show that the proposed approach achieves competitive or state-of-the-art performance while using significantly fewer trainable parameters compared to existing methods.

**Compliance With Llm Reviewing Policy:**

Affirmed.

**Final Justification:**

Thank you to the authors for their detailed responses. My concerns have been largely addressed. I hope the authors will actively incorporate these discussions into the final version to enhance the paper’s clarity and depth. I am raising the rating to Accept. Good luck.

**Key Questions For Authors:**

```
Q1. Could the authors clarify which aspects of the proposed design differ fundamentally from existing adapter-based tuning approaches, and provide additional comparisons or analyses to better highlight the methodological novelty?
Q2. The model appears to rely on frame sampling and simple aggregation without explicit temporal modeling. How does the proposed approach capture temporal distortions or motion-related artifacts that are critical for video quality assessment?
Q3. The appendix suggests that different datasets require specific fine-tuning strategies. Do existing VQA methods also rely on such dataset-specific tuning, and how sensitive is the proposed method to hyperparameter choices across datasets?
Q4. The paper introduces a five-level quality prompt mechanism. How sensitive is the model to the number of quality levels or the semantic initialization of these prompts, and have alternative prompt configurations been explored?
```

**Limitations:**

```
No. The discussion of limitations and potential societal impacts is minimal. The authors should provide a clearer discussion of methodological limitations, such as dependence on large pretrained models and limited temporal modeling, as well as potential dataset bias in subjective quality annotations.
```

**Strengths And Weaknesses:**

### Strengths
```
1. The paper has a clear motivation and aligns with current research trends in the field.
2. The proposed method is parameter-efficient and should be encouraged.
3. Figure 2-4 vividly illustrates the design details of the proposed method.
4. Extensive experiments on multiple VQA datasets support the work.
```

### Weaknesses
```
1. The paper's innovation appears to be primarily engineering-driven, introducing a parameter-efficient adapter. The methodological innovation is somewhat limited, as the integration of existing techniques such as adapter-based tuning and prompt learning has already been extensively studied. Authors are advised to strengthen the comparison and analysis of relevant technologies.
2. The paper appears to rely on frame sampling strategies and simple aggregation without explicitly modeling temporal dynamics. It also lacks an in-depth discussion on whether the model can capture critical temporal information in videos.
3. Section D.2 of the appendix indicates that the paper employs specific fine-tuning strategies for different datasets, which potentially suggests that the model may lack robustness across different distributions. The paper also lacks a detailed analysis of related hyperparameter sensitivity. Do existing methods also adopt such specific fine-tuning strategies?
4. Regarding a core quality level prompt mechanism in the paper, there appears to be no discussion of the impact of the number of levels or the initial semantic selection on performance. Furthermore, the paper lacks discussion of highly relevant vision-language models and quality-level prompt mechanisms in related quality assessment studies for video perception and understanding tasks. For example,
- Quality-guided vision-language learning for long-term action quality assessment, TMM, 2025.
- Vision-language action knowledge learning for semantic-aware action quality assessment, ECCV, 2024.
- Language-guided audio-visual learning for long-term sports assessment, CVPR, 2025.
```

---

> ### Author Rebuttal · Authors · 2026-03-29
>
> ### **We sincerely thank the reviewer for the positive and insightful comments. We are encouraged that the reviewer finds the paper well-motivated, clearly presented, parameter-efficient, and supported by extensive experiments.**
>
> ### **Question 1: Fundamental differences from existing adapter-based tuning.**
>
> ### **Answer：**
> We thank the reviewer for this helpful suggestion and agree that this distinction should be made clearer. Existing adapter-based tuning methods for CLIP are typically branch-local and mainly refine encoder outputs for downstream transfer. In contrast, SCMA is designed as a shared cross-modal adaptation mechanism: it is inserted into both visual and textual branches, shares bottleneck projections across modalities within each layer, and further shares the adapter core across layers. As a result, SCMA does not simply refine features independently, but adapts both modalities in a unified quality-aware joint space for similarity-based VQA, which is particularly important for fine-grained perceptual quality prediction. Empirically, Table 3 shows that SCMA outperforms representative tuning strategies such as CLIP-Adapter, CoOp, VPT, LoRA, and full fine-tuning, while Table 4 further confirms that the gains come progressively from dual-branch adaptation, cross-modal sharing, and inter-layer sharing. We will clarify this methodological distinction more explicitly in the revision.
>
>
> ### **Question 2: No explicit temporal modeling.**
>
> ### **Answer：**
> We appreciate the reviewer for this careful comment. **We would like to emphasize that this issue has already been explicitly studied in Appendix F.3 through dedicated experiments and analysis.** While Q-CLIP does not introduce a heavy explicit temporal head, this does not mean that temporal distortions or motion-related artifacts are ignored. As stated in the paper, we intentionally adopt a video-tuned CLIP backbone, since static-image CVLMs are not well equipped to model temporal dynamics; under this backbone, per-frame features already encode motion patterns and temporal consistency in a way that is compatible with mean pooling. More importantly, Appendix F.3 / Figure 10 directly compares mean pooling with Transformer- and Mamba-based temporal heads, and shows that mean pooling consistently performs best across all evaluated datasets. This suggests that adding trainable temporal modules may instead disrupt the pretrained representation space and hurt transfer. Therefore, the absence of an additional temporal head is a deliberate design choice rather than a limitation of the framework.
>
>
> ### **Question 3: Dataset-specific fine-tuning & cross-dataset hyperparameter sensitivity.**
>
> ### **Answer：**
> We thank the reviewer for this careful comment. **This issue has also been directly addressed in Appendix D.3.** In the main paper, we use slightly different fine-tuning settings on the five small datasets only to obtain a small additional gain. To verify that Q-CLIP does not rely on such dataset-specific tuning, we further evaluate a single unified fine-tuning strategy in Appendix D.3/Table 8, where the entire 0.14M-parameter SCMA is trained with the same learning rate and schedule across all datasets. This unified setting still achieves SOTA performance on all five small datasets, with results very close to those of the per-dataset strategy. These results show that Q-CLIP does not depend on dataset-specific tuning and is not highly sensitive to hyperparameter choices across datasets.
>
> ### **Question 4: Sensitivity to quality-level number and prompt semantic initialization.**
>
> ### **Answer：**
> We appreciate the reviewer for this important question. We respond from two aspects.
> - **Quality-level number:** In the paper, we already compare our five-level prompts with the widely used two-level antonym prompts in Table 5. The results consistently show that the five-level design performs better, indicating that a finer-grained quality hierarchy is more suitable for VQA than a coarse binary formulation.
> - **Prompt semantic initialization:** We further conduct additional experiments on LSVQ using several different prompt initializations, including “a \<level\>-quality video,” “a
> \<level\> video,” and “\<level\>-quality.” These variants lead to the same performance, showing that the model is not sensitive to the specific semantic initialization of the prompts. We believe this is mainly because our design includes learnable prompt tokens, which can effectively adapt the prompt representation during training and largely eliminate the bias introduced by different initial wordings. Therefore, our results suggest that the effectiveness of the prompt mechanism mainly comes from the quality-level design itself and the learnable adaptation, rather than from a particular handcrafted prompt initialization.

---

> > ### Author Rebuttal · Reviewer_Pxyv · 2026-04-02
> >
> > Thank you to the authors for their detailed responses. My concerns have been largely addressed. I hope the authors will actively incorporate these discussions into the final version to enhance the paper’s clarity and depth. I am raising the rating to Accept. Good luck.

---

### Official Review · Reviewer_xvQq · 2026-03-13

**Soundness:** 3
**Presentation:** 3
**Significance:** 3
**Originality:** 3
**Overall Recommendation:** 5
**Confidence:** 4

**Summary:**

Q-CLIP, a no-reference video quality assessment (VQA) method built on a frozen CLIP-style vision–language backbone. The authors introduce a Shared Cross-Modal Adapter (SCMA) to adapt the frozen encoder and use five learnable quality prompts to estimate video quality through similarity matching. The method also studies several frame sampling strategies and reports strong results on common VQA benchmarks such as LSVQ, KoNViD-1k, LIVE-VQC, YouTube-UGC, and CVD2014. Overall, the approach shows that a frozen CVLM can serve as the main backbone for VQA with only a small number of trainable parameters.

**Compliance With Llm Reviewing Policy:**

Affirmed.

**Key Questions For Authors:**

How much of the reported improvement comes from the SCMA adapter and quality prompt design, compared to simply using the stronger video-pretrained CLIP backbone?

The mixed sampling strategy averages predictions from multiple samplers at inference time. How does the model perform when using a single sampler under the same compute budget, without test-time ensembling?

Has the method been evaluated on newer VQA scenarios, such as HDR videos, AI-generated videos, or other emerging datasets? If not, how well do the authors expect the model to generalize to these settings?

**Limitations:**

The method still depends on a large frozen CLIP backbone, which may limit deployment in resource-constrained settings. The best results rely on test-time ensembling across multiple sampling strategies, increasing inference cost. In addition, the evaluation mainly covers traditional UGC VQA benchmarks

**Strengths And Weaknesses:**

The paper tackles an important problem in video quality assessment and explores the idea of using contrastive vision–language models directly as the VQA backbone, which is an interesting direction. The method is relatively simple and efficient, requiring only a small adapter module while keeping the main CLIP model frozen. The empirical results are strong and consistently outperform several previous CLIP-based VQA methods. The ablation studies are also helpful and show the contributions of the shared adapter, prompt design, and sampling strategies.
The method mainly combines known ideas such as prompt tuning and lightweight adapters on top of a pretrained CLIP backbone. The sampling strategy called “mixed sampling” effectively works as a test-time ensemble, which makes it difficult to determine how much improvement actually comes from the proposed sampling method itself. In addition, the improvements partly rely on a video-pretrained CLIP backbone, so it is unclear how much of the gain comes from the proposed design versus the stronger pretrained model. Finally, the evaluation mainly focuses on standard UGC VQA benchmarks and does not explore more recent directions such as MLLM-based VQA or HDR/AIGC video quality tasks, which are becoming important in the field.

---

> ### Author Rebuttal · Authors · 2026-03-29
>
> ### **We sincerely thank the reviewer for the positive and insightful comments. We are encouraged that the reviewer recognizes the importance of the problem, the interesting CVLMs-based direction, the efficiency of the lightweight design, and the strong empirical validation of the proposed method.**
>
> ### **Question 1: SCMA and quality prompt design vs. the stronger video-pretrained CLIP backbone.**
>
> ### **Answer：**
> We appreciate the reviewer’s comment, and we are glad to further clarify this point. **This question is already substantially addressed by the controlled experiments in our paper**, which are designed to disentangle the effects of the backbone, SCMA, and prompt design. Most directly, Table 6 shows that under the same experimental setup, Q-CLIP remains consistently strong with three standard image-pretrained CLIP backbones, while the stronger video-pretrained backbone brings only marginal additional gains. This indicates that the reported improvement does not mainly come from simply using a stronger backbone. Table 3 further shows that our proposed SCMA outperforms representative fine-tuning strategies, including full fine-tuning, CoOp, VPT, CLIP-Adapter, and LoRA, confirming that the gain comes from the proposed adaptation mechanism itself rather than generic backbone tuning. Table 5 then shows that our five-level prompt design is consistently better than the widely used antonym-based prompts, and in Appendix F.5/Table 13 the same prompt strategy also improves MaxVQA, indicating that the prompt design is generally effective rather than specific to Q-CLIP alone. Finally, Table 4 and Figure 5 provide additional support for SCMA by showing clear progressive gains from branch-wise adaptation, cross-modal sharing, and inter-layer sharing. Overall, these results consistently support that the reported improvement mainly comes from the proposed SCMA + prompt design, rather than simply from a stronger video-pretrained CLIP backbone.
>
> ### **Question 2: Contribution of the proposed sampling design vs. test-time ensembling & single-sampler performance under the same compute budget.**
>
> ### **Answer：**
> We thank the reviewer for this important question. **We would like to clarify that our goal is not to propose a standalone new sampling method, but to study how different sampling choices affect the performance of VQA.** We agree that mixed sampling introduces a test-time ensemble effect. However, this is only an optional enhancement, not the source of Q-CLIP’s strong performance. As shown in Table 1, single-sampler Q-CLIP already achieves SOTA performance, and as shown in Table 2, Q-CLIP still achieves SOTA results using only uniform sampling. To further address this concern, we additionally evaluated a no-ensemble setting under the same compute budget, where only one sampler is randomly selected at both training and inference. Q-CLIP remains at the SOTA level in this setting, although it is slightly inferior to the mixed-sampling that uses test-time ensembling. Therefore, mixed sampling provides an additional inference-time gain, but does not account for the core effectiveness of Q-CLIP.
> | Methods         | LSVQ$_{test}$ | LSVQ$_{1080p}$ |  KoNViD-1K  |   LIVE-VQC  |
> |-----------------|:-------------:|:--------------:|:-----------:|:-----------:|
> |    Mixed(w/)    |  0.899/0.900  |   0.823/0.866  | 0.896/0.901 | 0.826/0.867 |
> |    Mixed(w/o)   |  0.897/0.897  |   0.821/0.860  | 0.895/0.900 | 0.824/0.864 |
>
>
>
> ### **Question 3: Generalization to newer VQA datasets (HDR, AI-generated videos, and emerging datasets).**
>
> ### **Answer：**
> We thank the reviewer for this insightful comment. We would like to emphasize that, beyond the standard UGC datasets reported in the main paper, **we have already included additional experiments in the supplementary material on one recent UGC dataset (KVQ) and one HDR dataset (YouTube SFV+HDR)**. As shown in Appendix H, on both datasets, our method shows significant performance advantages over the best previously reported methods. This provides direct evidence that Q-CLIP generalizes well beyond the conventional settings in the main paper, and remains effective on both newer UGC data and HDR video data.
> For AI-generated video quality assessment, we agree that this is an important emerging direction. However, at present, we have not identified a public, complete, and readily usable AIGC VQA dataset that supports a fair and reproducible comparison. Therefore, we did not include this setting in the current submission. We will continue to closely follow this direction, and once a suitable public dataset becomes available, we will include it in our future experimental evaluation.

---

### Official Review · Reviewer_eA4S · 2026-03-13

**Soundness:** 3
**Presentation:** 4
**Significance:** 3
**Originality:** 3
**Overall Recommendation:** 5
**Confidence:** 4

**Summary:**

This paper proposes Q-CLIP, a fully CVLM-based framework for no-reference video quality assessment. The method adapts a frozen vision-language backbone using a lightweight Shared Cross-Modal Adapter (SCMA), introduces learnable five-level quality prompts, and studies frame-difference-based sampling for selecting informative video frames. The paper is clearly motivated by the limitations of classification-based pretraining for VQA and by the need for parameter-efficient adaptation. Extensive experimental results demonstrate that Q-CLIP offers significant performance and efficiency advantages over the latest SOTA models.

**Compliance With Llm Reviewing Policy:**

Affirmed.

**Final Justification:**

The authors addressed my concerns during the rebuttal. I keep my already positive (accept) recommendation.

**Key Questions For Authors:**

Please see weakness.

**Limitations:**

Please see weakness.

**Strengths And Weaknesses:**

Strengths
1. This study is highly innovative and well-motivated. It investigates how to adapt CVLMs to VQA tasks with minimal additional training overhead, while also systematically examining the effect of different frame sampling strategies on VQA. Overall, the paper offers valuable insights and provides a useful foundation for future research in this area.
2. This manuscript is well-written, and the figures and tables are presented clearly. And the proposed SCMA is simple, clean, and parameter-efficient, with cross-modal sharing and inter-layer sharing that are well aligned with the goal of lightweight adaptation.
3. The experimental section is comprehensive and thorough. Compared to current state-of-the-art models, Q-CLIP not only demonstrates significant performance advantages but also maintains efficiency advantages.
4. The method is especially compelling from an efficiency perspective, requiring only 0.14M trainable parameters while outperforming substantially heavier alternatives.

Weakness
1. Certain design decisions could be explained more clearly; for example, why was the uniform sampling strategy chosen for the fine-tuning phase rather than the mixed sampling strategy, given that the mixed sampling strategy outperformed the uniform sampling strategy in the pre-training experiments?
2. In the ablation experiments for E-SCMA shown in Figure 5, the 12-layer model performs better. Why does the paper default to the results from the 6-layer model? If this was done to balance efficiency, the authors should provide a detailed explanation in the paper.

---

> ### Author Rebuttal · Authors · 2026-03-29
>
> ### **We sincerely thank the reviewer for the positive and insightful comments. We are encouraged that the reviewer finds the paper novel, well-motivated, and experimentally thorough, with strong efficiency advantages.**
>
> ### **Question 1: Why do we use uniform sampling during fine-tuning instead of mixed sampling?**
>
> ### **Answer：**
>
> Thank you for pointing this out. The key point is that, in our paper, **the study of sampling strategies is intended as an exploratory analysis, whereas the downstream fine-tuning experiments are mainly designed to evaluate the effectiveness of the Q-CLIP architecture itself.** For this reason, we conduct the sampling comparison in Table 1 on **LSVQ, the largest and most representative dataset** in our experiments. Its scale, together with both intra-dataset and cross-dataset evaluation, **makes it an appropriate testbed for assessing the behavior and generalization of different sampling strategies.** Although mixed sampling achieves the best overall results in this study, we regard this as evidence from the sampling analysis itself rather than as a default setting that must be used in every subsequent experiment.
>
> The purpose of Table 2 is different. There, **we aim to show that Q-CLIP can already achieve strong downstream performance on small datasets under a simple and controlled setting.** We therefore intentionally adopt uniform sampling during fine-tuning, so that the performance gain can be attributed more directly to the proposed architecture rather than being jointly affected by an additional sampling enhancement. Since the effect of different sampling strategies has already been systematically examined on LSVQ, repeating mixed sampling on much smaller datasets would add limited new evidence while making the interpretation less clean. Importantly, even under this simple setting, Q-CLIP still achieves SOTA performance on all five small datasets. We will clarify this motivation more explicitly in the revised paper.
>
> ### **Question 2: Why is 6-layer E-SCMA the default instead of 12-layer E-SCMA?**
>
> ### **Answer：**
>
> Thank you for this helpful comment. **Figure 5 is intended to demonstrate the scalability of E-SCMA, rather than to suggest that the deepest configuration should be the default.**
>
> Our choice of 6 E-SCMA layers is aligned with the central goal of the paper: lightweight and parameter-efficient adaptation of frozen CVLMs. In Q-CLIP, SCMA is the only trainable component, and the default configuration uses only 0.14M trainable parameters. Although Figure 5 shows that increasing the number of E-SCMA layers can further improve performance, the 6-layer setting better reflects the main message of the paper, namely that strong VQA performance can already be achieved with an extremely small adaptation budget. Importantly, all comparative experiments in the paper use this 6-layer configuration, and it is already sufficient to achieve SOTA results on the downstream datasets.
>
> At the same time, we include deeper settings in Figure 5 to show that the proposed design is scalable and still has additional performance headroom. We will revise the paper to make this choice clearer, namely that the 6-layer setting is used as the default because it better balances accuracy, efficiency, and the lightweight design objective, while the 12-layer result is included to demonstrate the extensibility of SCMA rather than the preferred default configuration.

---

> > ### Author Rebuttal · Reviewer_eA4S · 2026-04-01
> >
> > My concerns are addressed.

---

### Decision · Program_Chairs · 2026-04-30

**Decision:**

Accept (regular)

**Comment:**

This paper proposes Q-CLIP, a parameter-efficient framework that adapts contrastive VLMs for video quality assessment through a shared cross-modal adapter and learnable quality prompts. The method demonstrates good performance across multiple benchmarks while maintaining very low training cost.

The paper received consistently strong positive feedback from all reviewers. Reviewers agreed that the work is well-motivated, technically sound, and supported by comprehensive experiments. In particular, the proposed SCMA design and prompt mechanism are considered effective and well-aligned with the goal of lightweight adaptation, and the empirical results show clear advantages in both performance and efficiency.

During the rebuttal phase, the authors provided detailed clarifications and additional experiments addressing concerns about the contribution of the backbone versus the proposed modules, the role of sampling strategies, and the generalization capability of the method. Reviewers confirmed that their concerns have been largely or fully resolved.

Overall, this is a solid and well-executed paper with clear contributions and practical relevance. The AC recommends acceptance.